# Understanding and Mitigating Distribution Shifts for Machine Learning Force Fields

## Abstract

Machine Learning Force Fields (MLFFs) are a promising alternative to expensive *ab initio* quantum mechanical molecular simulations. Given the diversity of chemical spaces that are of interest and the cost of generating new data, it is important to understand how MLFFs generalize beyond their training distributions. In order to characterize and better understand distribution shifts in MLFFs, we conduct diagnostic experiments on chemical datasets, revealing common shifts that pose significant challenges, even for large foundation models trained on extensive data. Based on these observations, we hypothesize that current supervised training methods inadequately regularize MLFFs, resulting in overfitting and learning poor representations of out-of-distribution systems. We then propose two new methods as initial steps for mitigating distribution shifts for MLFFs. Our methods focus on test-time refinement strategies that incur minimal computational cost and do not use *ab initio* labels. The first strategy, based on spectral graph theory, modifies the edges of test graphs to align with graph structures seen during training. It can be applied to any existing pre-trained model to mitigate connectivity distribution shifts. Our second strategy improves representations for out-of-distribution systems at test-time by taking gradient steps using an auxiliary objective. Inspired by previous test-time training works in computer vision, we replace self-supervised objectives at test time with an objective that uses an efficient prior to address distribution shifts. Our test-time refinement strategies can reduce force errors by an order of magnitude on out-of-distribution systems, suggesting that MLFFs are capable of and can move towards modeling diverse chemical spaces, but are not being effectively trained to do so. Our experiments establish clear benchmarks for evaluating the generalization capabilities of the next generation of MLFFs.

## 1 Introduction

Understanding the quantum mechanical properties of atomistic systems is crucial for the discovery and development of new molecules and materials. Computational methods like Density Functional Theory (DFT) are essential for studying these systems, but the high computational demands of such methods limit their scalability. Machine Learning Force Fields (MLFFs) have emerged as a promising alternative, learning to predict energies and forces from reference quantum mechanical calculations. MLFFs are faster than traditional *ab initio* methods, and their accuracy is rapidly improving for modeling complex atomistic systems (Batzner et al., 2022; Schütt et al., 2017; Gasteiger et al., 2021; Batatia et al., 2022).

Given the computational expense of *ab initio* simulations for all chemical spaces of interest, there has been a push to train larger and more accurate MLFFs, designed to work well across many different systems. A goal here is developing models with general representations that accurately capture diverse chemistries and eliminate the need to recollect data and retrain a model for each new system. To determine which systems an MLFF can accurately describe and to assess the reliability of its predictions, it is important to understand how MLFFs generalize beyond their training distributions. This understanding is essential for applying MLFFs to new and diverse chemical spaces, ensuring that they perform well not only on the data they were trained on, but also on unseen, potentially more complex systems.

We conduct an in-depth exploration to identify and understand distribution shifts. On example chemical datasets, we find that state-of-the-art models struggle with common distribution shifts (Kovács et al., 2023; Shoghi et al., 2023; Liao et al., 2024; Batatia et al., 2024) (see §2). These generalization challenges suggest that current supervised training methods for MLFFs overfit to training distributions and do not enable MLFFs to generalize accurately. We demonstrate that there are multiple reasons that this is the case,

**Figure 1: Distribution Shifts for MLFFs.** We visualize distribution shifts based on changes in features, labels, and graph structure. Typical training samples from SPICE Eastman et al. (2023) and new systems from SPICEv2 (Eastman et al., 2024) are displayed. A feature shift, such as a change in elements, is shown by replacing a carbon atom with a silicon atom (left). A force norm shift is shown by the close proximity of an $H_2$ molecule (circled in pink), leading to high force norms (middle). A connectivity shift is shown by the tetrahedral geometry in $P_4S_6$, which differs from the typical planar geometry seen during training (right).

including challenges associated with poorly-connected graphs and learning unregularized representations, evidenced by jagged predicted potential energy surfaces for out-of-distribution systems.

Building off of our observations, we propose two paths forward that take initial steps at mitigating distribution shifts for MLFFs without reference labels through test-time radius refinement and test-time training (Sun et al., 2020; Gandelsman et al., 2022; Jang et al., 2023). For test-time radius refinement, we modify the construction of test-graphs to match the training Laplacian spectrum, overcoming differences between training and testing graph structures. For test-time training (TTT), we address distribution shifts by taking gradient steps on an auxiliary objective at test time. Analogous to self-supervised objectives in computer vision TTT works (Gandelsman et al., 2022; Sun et al., 2020; Hardt & Sun, 2024), we use an efficient prior as a target to improve representations at test time.

We extensively test both approaches and show that our test-time refinement strategies are effective in mitigating distribution shifts for MLFFs. Our experiments establish clear benchmarks that highlight ambitious but important generalization goals for the next generation of MLFFs.

We summarize our main contributions here:

1. We run diagnostic experiments on different chemical datasets to characterize and understand common distribution shifts for MLFFs in §2.

2. Based on (1), we take first steps at mitigating MLFF distribution shifts in §3 with two test-time refinement strategies that mitigate these distribution shifts and improve accuracy. Note that we do not have access to *ab initio* test labels.

3. The success of these methods, validated through extensive experiments in §4, suggests that MLFFs are not being adequately trained to generalize, despite current models being expressive enough to tackle the distribution shifts explored in §2.

**Related Work.** There is rich literature studying distribution shifts in machine learning (Zhao et al., 2022; Sugiyama et al., 2007; Taori et al., 2020). We are also inspired by previous test-time training works (Sun et al., 2020; Gandelsman et al., 2022; Hardt & Sun, 2024; Jang et al., 2023). Previous work has also explored multi-fidelity MLFFs (Ramakrishnan et al., 2015; Giselle Fernández-Godino, 2023); our use of multiple fidelities differs in the NN training strategies used and is focused on understanding and mitigating distribution shifts. Finally, other work has begun finding systematic difficulties with generalization for MLFFs (Deng et al., 2024). For a more detailed discussion, see §A.

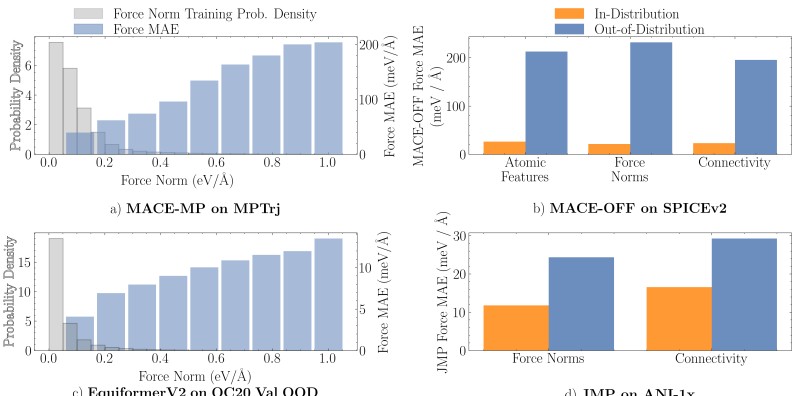

Figure 2: **Distribution Shifts for Large Foundation Models.** We study distribution shifts in the context of large foundation models that are designed for broad chemical spaces. (a) MACE-MP is a 15M parameter materials foundation model trained on 1.4M structures from MPTrj. We evaluate it on the MPTrj train set. (b) MACE-OFF is a large biomolecules foundation model trained on 951k structures from SPICE. We evaluate MACE-OFF on 10k new molecules from SPICEv2. (c) EquiformerV2 is a 150M parameter model trained on 100M+ structures from OC20. We evaluate it on the OC20 out-of-distribution validation set. (d) JMP is a 240M parameter model trained on 100M+ samples from OC20, OC22, ANI-1x, and Transition-1x. We evaluate JMP on the ANI-1x test set. A molecule is considered out-of-distribution if it is more than 1 standard deviation away from the mean training force norm or connectivity (with respect to the eigenvalue heuristic described in §2.2), or if it contains a new element. Despite their scale, these large foundation models have $2-10\times$ larger force mean absolute errors (MAE) when encountering distribution shifts.

## 2 DISTRIBUTION SHIFTS FOR MACHINE LEARNING FORCE FIELDS

### 2.1 PROBLEM SETUP AND BACKGROUND

MLFFs approximate molecule-level energies and atom-wise forces for a chemical structure by learning neural network parameters from data. For a given a molecular structure, the input to the ML model consists of two vectors: $\mathbf{r} \in \mathbb{R}^{n\times3}$, $\mathbf{z} \in \mathbb{R}^{n\times d}$, where $n$ represents the number of atoms in the molecule, $\mathbf{r}$ are the atomic positions, and $\mathbf{z}$ are the features of the atom, such as atomic numbers or whether an atom is fixed or not. The model outputs $\hat{E} \in \mathbb{R}, \hat{\mathbf{F}} \in \mathbb{R}^{n\times3}$, which are the predicted total potential energy of the molecule and the predicted forces acting on each atom. The learning objective is typically formulated as a supervised loss function, which measures the discrepancy between the predicted energies and forces and reference energies and forces:

$$\mathcal{L}(\mathbf{F},E) = \lambda_E ||E_{ref} - \hat{E}||_2^2 + \lambda_F \sum_{i=1}^{n} ||\mathbf{F_{i,ref}} - \hat{\mathbf{F_i}}||_2^2, \tag{1}$$

where $\lambda_E, \lambda_F$ are hyperparameters.

Most modern MLFFs are implemented as graph neural networks (GNNs) Gilmer et al. (2017). Consequently, $\hat{E}$ and $\hat{\mathbf{F}}$ are functions of $\mathbf{z}$, $\mathbf{r}$, and $A \in \mathbb{R}^{n\times n}$, the adjacency matrix representing the molecule:

$$\hat{E}, \hat{\mathbf{F}} = f(\mathbf{z}, \mathbf{r}, A) \tag{2}$$

The atoms in the molecule are modeled as nodes in a graph, and edges are specified by the adjacency matrix that includes connections to all atoms within a specified radius cutoff (Gasteiger et al., 2021; Batatia et al., 2022). The adjacency matrix fully determines a graph structure, and thus defines the graph over which the GNN performs its computation.

### 2.2 CRITERIA FOR IDENTIFYING DISTRIBUTION SHIFTS

In this section, we formalize criteria for identifying distribution shifts based on the features, labels, and graph structures in chemical datasets. We define these distribution shifts broadly to encompass the diversity of chemical spaces. We also note that distribution shifts can occur independently along each dimension: e.g., a shift in features does not necessarily imply a shift in labels (see §E for details). This categorization provides a framework for understanding the types of distribution shifts an MLFF may encounter (see Fig. 1). This understanding motivates the refinement strategies described in §3 that take first steps at mitigating these shifts, providing insights into why MLFFs are susceptible to these shifts in the first place.

**Distribution Shifts in Atomic Features (z).**   Distribution shifts in atomic features $\mathbf{z}$ are the most apparent and detrimental to the performance of current state-of-the-art models (see §4). This may involve encountering a molecule with a new element at test time that was not present during training. For example, a model trained on $CO_2$ might be tested on $SiO_2$ without having seen $Si$ during training (see Fig. 1). Although this might initially seem like an unreasonably hard task, we argue that a truly general machine learning model for quantum chemistry should be capable of handling arbitrary elements and charges.

**Distribution Shifts in Forces (F).**   An MLFF may also encounter a distribution shift in the force labels it predicts. A model trained on structures close to equilibrium, with low force magnitudes, might be tested on a structure with higher force norms. Fig. 1 shows an example of a tightly clustered $H_2$ molecule, which leads to a force norm distribution shift.

**Distribution Shifts in Graph Structure and Connectivity ($A$).**   Since many MLFFs are implemented as GNNs, they may encounter distribution shifts in the graph structure defined by $A$. We refer to these as connectivity distribution shifts because $A$ determines the graph connectivity used by the GNN. Connectivity distribution shifts are particularly common in molecular datasets, where one could encounter a benzene ring at test time, despite only having trained on long acyclic structures. Fig. 1 provides an example of a connectivity distribution shift, going from planar training structures to a tetrahedral geometry at test time.

We identify connectivity distribution shifts by analyzing the eigenvalue spectra of the graph Laplacian:

$$L = I - (D)^{-\frac{1}{2}} A (D)^{-\frac{1}{2}}, \tag{3}$$

where $D \in \mathbb{R}^{n \times n}$ is the degree matrix ($D_{ii} = \text{degree}(\text{node}_i)$ and $D_{ij} = 0$ for $i \neq j$, $A_{ij} = 1$ if $||r_i - r_j||_2 \leq r_{cutoff}$ and $0$ otherwise), and $I$ is the identity. $L$ has eigenvalues $\lambda_0, \leq \lambda_1, \leq \cdots \leq \lambda_{n-1}$, where $\lambda_i \in [0,2] \, \forall i$, and the multiplicity of the $0$ eigenvalue equals the number of connected components in the graph.

By comparing the eigenvalue distributions between two graphs, we determine whether they are structurally different. While it's possible to use a divergence measure over the eigenvalue distributions to characterize the difference, we found that looking at the difference in the clustering around the 1 eigenvalue between graphs serves as a practical heuristic. Most molecular datasets we studied had structures with a consistent fraction of the eigenvalues clustered around 1, indicating mostly regular and well-connected graphs (see §3.1).

**Observed Distribution Shifts for Large Foundation Models.**   We contextualize the aforementioned distribution shifts by considering four large foundation models: MACE-OFF, MACE-MP, EquiformerV2, and JMP (Kovács et al., 2023; Shoghi et al., 2023; Liao et al., 2024; Batatia et al., 2024) MACE-OFF is a 4.7M biomolecules foundation model trained on 951k structures primarily from the SPICE dataset (Eastman et al., 2023). The 15M parameter MACE-MP foundation model is trained on 1.5M structures from the Materials Project (Deng, 2023). EquiformerV2 is a 150M parameter model trained on 100M+ structures from OC20 (Chanussot et al., 2021). The JMP model has 240M parameters and is trained on 100M+ structures from OC20, OC22, ANI-1x, and Transition-1x (Chanussot et al., 2021; Tran et al., 2023; Smith et al., 2020; Schreiner et al., 2022). These models represent four of the largest open-source MLFFs to date, and they have been trained on some of the most extensive datasets available. We focus on these models since their scale is designed for tackling broad chemical spaces.

We examine the generalization ability of MACE-OFF by testing it on 10k new molecules from the SPICEv2 dataset (Eastman et al., 2024) not included in the MACE-OFF training set. A molecule is defined as out-of-distribution if it is more than 1 standard deviation away from the mean training data force norm or connectivity (with respect to the eigenvalue heuristic defined above §2.2), or if it contains a new element. Despite its scale, MACE-OFF performs worse by an order of magnitude on out-of-distribution systems (see Fig. 2b).

We evaluate JMP on the ANI-1x (Smith et al., 2020) test set defined in Shoghi et al. (2023). Although this test set does not have new elements, JMP also suffers predictably from force norm and connectivity distribution shifts (see Fig. 2d).

We focus on force norm distribution shifts for MACE-MP and EquiformerV2, since connectivity is more uniform across bulk materials and catalysts, where atoms are packed tightly into a periodic cell. For MACE-MP, we evaluate its performance directly on the entire MPTrj dataset. This model does not have a clear validation set, as it was trained on all of the data to maximize performance (Batatia et al., 2024). MACE-MP still clearly performs worse as force norms deviate from the majority of the training distribution (see Fig. 2a). The performance deterioration would be more severe with a held-out test set. EquiformerV2 also struggles with high force norm structures when evaluated on the validation out-of-distribution set from OC20 (Chanussot et al., 2021) (see Fig. 2c).

**Observations.** Training larger models with more data (for example with active learning (Vandermause et al., 2020)) is one approach to address these distribution shifts. However, doing so can be computationally expensive. Our diagnostic experiments also indicate that scale alone might not fully address distribution shifts, as naively adding more in-distribution data does not help large models generalize better (see Fig. 2). The diversity of chemical spaces makes it exceedingly difficult to know the exact systems that an MLFF will be tested on *a priori*, making it challenging to curate the perfect training set. These observations lead us to develop strategies that mitigate distribution shifts by modifying the training and testing procedure of MLFFs. Importantly, these refinement strategies can be combined with any further architecture and data advances.

## 3 Understanding and Mitigating Distribution Shifts with Test-Time Refinement Strategies for MLFFs

Based on the generalization challenges for the large foundation models (see §2), we hypothesize that many MLFFs are severely overfitting to the training data, resulting in a failure to learn generalizable representations. Building on our observations in §2 and to test this hypothesis, we develop two test-time refinement strategies that also mitigate distribution shifts. We focus on test time evaluations, i.e., with access to test molecular structures but without access to reference labels. First, by studying the graph Laplacian spectrum, we investigate how MLFFs, and GNNs in general (Bechler-Speicher et al., 2024), tend to overfit to the regular and well-connected training graphs. In §3.1, we address connectivity distribution shifts by aligning the Laplacian eigenvalues of a test structure with the connectivities of the training distribution. Second, we show that MLFFs are inadequately regularized, resulting in poor representations of out-of-distribution systems. We incorporate inductive biases from a cheap physical prior using our pre-training and test-time training procedure (§3.2) to regularize the model and learn more general representations, evidenced by smoother predicted potential energy surfaces. The effectiveness of these test-time refinement strategies, validated through extensive experiments in §4 and §C, may indicate that MLFFs are currently poorly regularized and overfit to graph structures seen during training, hindering broader generalization.

### 3.1 Test-Time Radius Refinement

We hypothesize that MLFFs tend to overfit to the specific graph structures encountered during training. We can characterize graph structures by studying the Laplacian spectrum of a graph. At test time, we can then identify when an MLFF encounters a graph with a Laplacian eigenvalue distribution that significantly differs from the training graphs (see 2.2). To address this shift, we propose updating the test graph to more closely resemble the training graphs, thereby mitigating connectivity distribution shifts. Since the adjacency matrix $A$ and graph Laplacian $L$ are typically generated by a radius graph, we refine the radius cutoff at test time. Instead of using a fixed radius cutoff $r_{\text{train}}$ for both training and testing, adjusting the radius cutoff at test time can help achieve a connectivity that more closely resembles the training graphs.

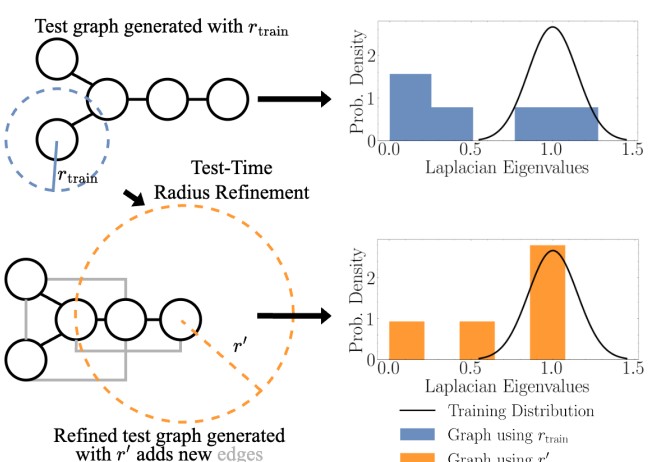

Figure 3: **Test-Time Radius Refinement.** MLFFs tend to overfit to the well-connected graphs seen during training, which can be identified by the clustering of Laplacian eigenvalues around 1. To mitigate connectivity distribution shifts at test time, we find the optimal radius cutoff, which aligns the Laplacian eigenvalues of test graphs with those of the training distribution.

Formally, for each test structure $j$, we search over $k$ new radius cutoffs $[r_i]_{i=1}^k$, calculate the new eigenvalue spectra for $L^{(j)}$ induced by the new cutoff $r_i$, and select the $r_i$ that minimizes the difference between the eigenvalue spectra of the new graph and the training graphs (see Fig. 3):

$$r_{\text{test}}^{(j)} = \underset{[r_i]_{i=1}^k}{\arg\min} D(\lambda_{\text{train}}, \lambda(L^{(j)}(r_i))), \tag{4}$$

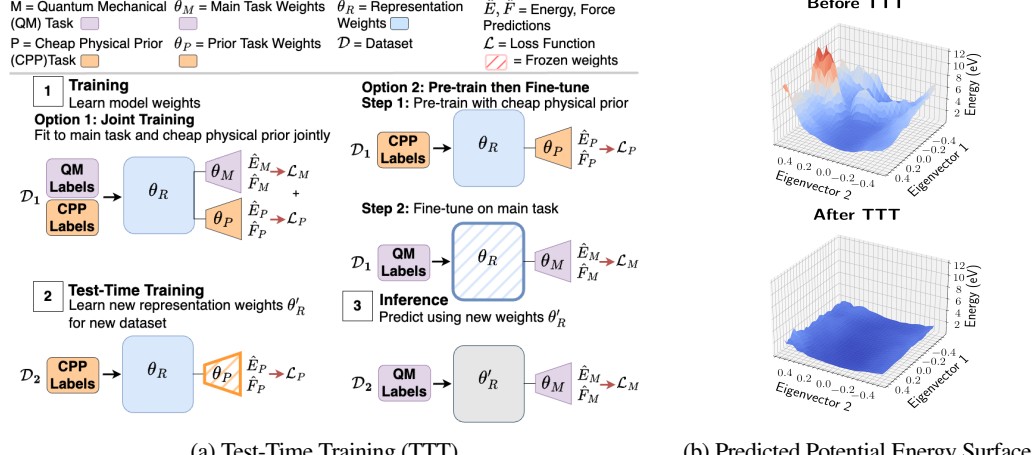

(a) Test-Time Training (TTT)      (b) Predicted Potential Energy Surface

Figure 4: **Test-Time Training Mitigates Distribution Shifts and Smooths Predicted Potential Energy Surfaces.** We hypothesize that due to overfitting, the predicted potential energy surfaces are jagged for out-of-distribution systems. Our proposed test-time training method (TTT, a) regularizes MLFFs by incorporating inductive biases into the model using a cheap prior. Test-time training first learns useful representations from the prior using either joint-training or a pre-train, freeze, and fine-tune approach. TTT then updates the representations at test-time using the prior to improve performance on out-of-distribution samples. We plot the predicted potential energy surface from a GemNet-dT model along the 2 principal components of the Hessian for salicylic acid, a molecule not seen during training, before and after test-time training (b). TTT effectively smooths the potential energy landscape and improves errors.

where $\lambda_{\text{train}}$ is the training distribution of eigenvalues, $\lambda(L^{(j)}(r_i))$ is the eigenvalues distribution of the Laplacian matrix for sample $j$ generated with radius cutoff $r_i$, and $D$ is some distance function.

As a heuristic for the distance function $D$, we characterize eigenvalue distributions by calculating the percentage of eigenvalues that fall within the range of $[0.9, 1.1]$, which roughly corresponds to the mean training Laplacian eigenvalue $\pm$ half the standard deviation for common molecular datasets. For many molecular datasets (such as SPICE, MD17, and MD22), we found that this percentage is consistently around $55\%$ (Chmiela et al., 2023; 2017; Eastman et al., 2023; Smith et al., 2020). Consequently, we implement test-time radius refinement by selecting a new radius cutoff $r'$ from $k = 10$ candidates that yields a distribution with approximately $55\%$ of its eigenvalues clustered close to 1. We leave the investigation of other distance metrics to future work. For further details and theoretical motivation, see §F.

We emphasize that test-time radius refinement can be applied to any existing MLFF that uses a radius graph, including existing foundation models like MACE-OFF (Kovács et al., 2023) and JMP (Shoghi et al., 2023). This approach incurs minimal computational cost and can be done quickly even on a CPU, requiring only eigenvalue computations. Our experiments show that this procedure virtually never deteriorates performance, as one can always revert to the same radius cutoff used during training (see §4). This refinement method addresses the source of connectivity distribution shifts and serves as a simple and effective initial strategy for handling new connectivities.

### 3.2 TEST-TIME TRAINING USING CHEAP PRIORS

We further hypothesize that the current supervised training procedure for MLFFs can lead to overfitting, leading to poor representations for out-of-distribution systems and jagged potential energy landscape predictions (see Fig. 4b for an example on salicylic acid (Chmiela et al., 2017)). To address this, we propose introducing inductive biases through improved training and inference strategies to smooth the predicted energy surfaces. The smoother energy landscape from the improved training indicates that the model may have learned more robust representations, mitigating force norm, element, and connectivity distribution shifts.

We represent these inductive biases as cheap priors, such as classical force fields or simple ML models. These priors can evaluate thousands of structures per second using only a CPU, making them computationally efficient for test-time use. First, we describe our pre-training procedure, which ensures the MLFF learns useful representations from the cheap prior. By leveraging these representations, we can smooth the predicted energy landscape and mitigate distribution shifts by taking gradient steps with our test-time training (TTT) procedure.

**Pre-Training with Cheap Physical Priors.** We propose a training strategy that first pre-trains on energy and force targets from a cheap prior and then fine-tunes the model on the ground truth quantum mechanical labels. Our loss function for one structure is defined as:

$$\mathcal{L}(\mathbf{F}^M, E^M, \mathbf{F}^P, E^P) = \mathcal{L}_M + \mathcal{L}_P = \sum_{l \in \{M,P\}} \left( \lambda_{E^l} ||E^l - \hat{E}^l||_2^2 + \lambda_{F^l} \sum_{i=1}^{n} ||\mathbf{F_i^l} - \hat{\mathbf{F}_i^l}||_2^2 \right), \tag{5}$$

where $\hat{E}, \hat{\mathbf{F}}$ are the predicted energy and forces, and $M$ and $P$ denote the main and prior task, respectively. During pre-training, gradient steps are initially only taken on the prior objective, corresponding to $\mathcal{L}_P$. For fine-tuning, the representation parameters, $\theta_R$, learnt from the prior are kept frozen, and the main task parameters, $\theta_M$, are updated by training only on the main task loss, $\mathcal{L}_M$. Pre-training and fine-tuning can also be merged and the model can be *jointly trained* on both the cheap prior targets and the expensive DFT targets (see Fig. 4a). This corresponds to training on $\mathcal{L}_P + \mathcal{L}_M$. Freezing or joint-training both force the main task head to rely on features learnt from the prior. This approach acts as a form of regularization, resulting in more robust representations. It enables the prior to be used to improve the features extracted from an out-of-distribution sample at test time, improving main task performance. For more details on the necessity of proper pre-training for test-time training, see §B.

**TTT Implementation Details.** For clarity, let us separate our full model into its three components: $g_{\theta_R}$ (the representation model), $h_{\theta_M}$ (the main task head), and $h_{\theta_P}$ (the prior task head). The representation parameters, $\theta_R$, are learned by minimizing $\mathcal{L}$ during joint training (see Eq. 5), or by minimizing $\mathcal{L}_P$ during pre-training and then freezing them during the fine-tuning phase. Test-time training involves the following steps:

1. **Updating representation parameters.** At test-time, we update $\theta_R$ by minimizing the prior loss, $\mathcal{L}_P$, on samples from the test distribution $\mathcal{D}_{test}$, which are labeled by the cheap prior. This is expressed as:

$$\theta'_R = \underset{\theta_R}{\operatorname{argmin}} \mathbb{E}_{(\mathbf{r},\mathbf{z},\mathbf{F^P},E^p) \sim \mathcal{D}_{test}} [\mathcal{L}_P(h_{\theta_P} \circ g_{\theta_R}(\mathbf{r},\mathbf{z}), \mathbf{F^P}, E^p)]. \tag{6}$$

During this process, the prior head parameters, $\theta_P$, are kept frozen during test-time updates. This incorporates inductive biases about the out-of-distribution samples into the model, regularizing the energy landscape and helping the model generalize (see Fig. 4b and Fig. 16).

2. **Prediction on test set.** Once the representation parameters are updated, we predict the main task labels for the test set using the newly adjusted representation:

$$\hat{E}, \hat{\mathbf{F}} = h_{\theta_M} \circ g_{\theta'_R}(\mathbf{r}, \mathbf{z}). \tag{7}$$

The parameters $\theta'_R$ are recalculated with Eq. 6 when a new out-of-distribution region is encountered (i.e., when testing is done on a new system). See Fig. 4a for an outline of our method.

We emphasize that TTT only uses the prior labels at test time, and these priors are efficient to compute (see §G). Priors are also widely available, since one can always use a simple analytical potential (like Lennard-Jones (Schwerdtfeger & Wales, 2024)), or widely applicable semi-empirical potential (Bannwarth et al., 2019). We reiterate that TTT with a prior will not work on a model that has only been trained on reference calculations. The main task head must use the features learnt from the prior so that when those features are improved with the prior at test time, the main task performance will also improve (see §B).

## 4 EXPERIMENTS

We conduct experiments on chemical datasets to both identify the presence of distribution shifts and evaluate the effectiveness of our test-time refinement strategies to mitigate these shifts. In §4.1, we find distribution shifts on the SPICE dataset with the MACE-OFF foundation model (Eastman et al., 2023; Kovács et al., 2023). In §4.2, we explore extreme distribution shifts and demonstrate that our test-time refinement strategy enables stable simulations on new molecules, even when trained on a limited dataset of 3 molecules from the MD17 dataset (Chmiela et al., 2017). Finally, in §C.4, we assess how our test-time refinement strategy can handle high force norms in the MD22 dataset when the model is trained only on low force norms. Although matching in-distribution performance remains a challenging open machine learning problem (Sun et al., 2020; Gandelsman et al., 2022), our experiments indicate that test-time refinement strategies are a promising initial step for addressing distribution shifts with MLFFs. The improvements from these test-time refinement strategies also suggest that MLFFs can be trained to learn more general representations that are resilient to distribution shifts. Additional experiments with more models, datasets, and priors are provided in §C.

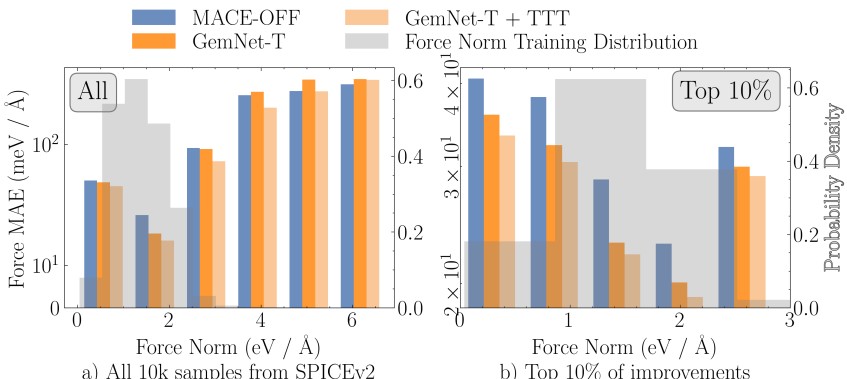

Figure 5: **Evaluating Distribution Shifts for Force Norms on SPICEv2.** The MACE-OFF model is trained on 951k samples from the SPICE dataset, with the training force norm distribution shown in gray. We evaluate MACE-OFF on new molecules from the SPICEv2 dataset with varying force norms. As the force norms deviate further from the training distribution, MACE-OFF's force errors increase. We also train a GemNet-T, and then apply test-time training (TTT), mitigating this shift. We highlight the top 10% of molecules with the greatest improvement to demonstrate that TTT is effective even for structures that are near the training distribution in (b).

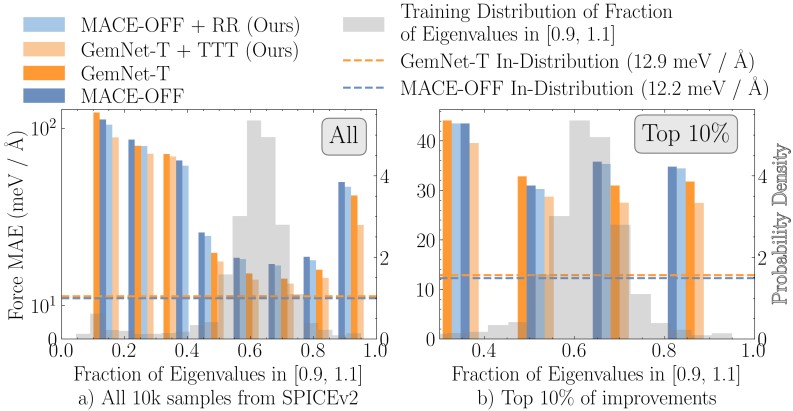

Figure 6: **Evaluating Connectivity Distribution Shifts on SPICEv2.** The majority of the MACE-OFF model's training structures have about 55% of their Laplacian eigenvalues in $[0.9, 1.1]$ (shown in gray). We evaluate MACE-OFF on new molecules from the SPICEv2 dataset with varying connectivity, defined by the Laplacian eigenvalue heuristic (see §3.1 for details). Test structures with different connectivity incur larger errors for MACE-OFF. Test-time training (TTT) applied to a GemNet-T model and test-time radius refinement (RR) applied to MACE-OFF are both able to mitigate this performance drop at minimal computational cost. We highlight the top 10% of molecules with the greatest improvement to demonstrate that TTT is effective even for connectivities close to the training distribution in (b).

## 4.1 EVALUATING DISTRIBUTION SHIFTS IN MACE-OFF: FROM SPICE TO SPICEV2

We investigate distribution shifts from the SPICE dataset to the SPICEv2 dataset (Eastman et al., 2023; 2024) by analyzing the MACE-OFF foundation model (Kovács et al., 2023). As shown in Fig. 5, Fig. 6, and Fig. 7, we observe that despite being trained on 951k data points and scaled to 4.7M parameters, MACE-OFF experiences force norm, connectivity, and element distribution shifts when evaluated on 10k new molecules from SPICEv2 (Eastman et al., 2024). Any deviation from the training distribution, shown in gray, results in an increase in force error.

We evaluate the effectiveness of our test-time refinement strategies in mitigating these distribution shifts. For the MACE-OFF model, we implement test-time radius refinement (RR) by searching over 10 different radius cutoffs and selecting the one that best matches the training Laplacian eigenvalue distribution (see §3.1). We also train a GemNet-T model on the same training data used by MACE-OFF, using the pre-training, freezing and fine-tuning method described in §3.2, with the sGDML model as the prior (Chmiela et al., 2019). See D for more details.

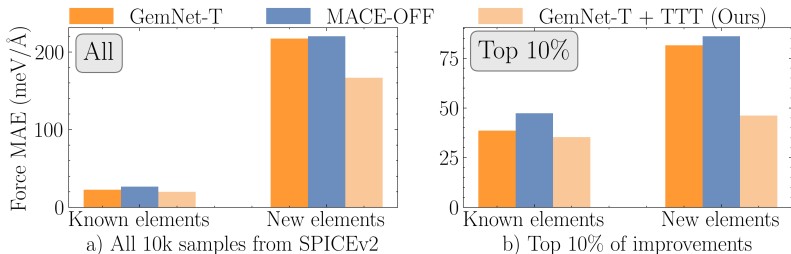

Figure 7: **Evaluating New Elements Distribution Shift on SPICEv2.** The MACE-OFF model deteriorates in performance when encountering new elements in the SPICEv2 dataset. A GemNet-T model is able to mitigate this shift with TTT. We highlight the top 10% of molecules with the greatest improvement, showing that TTT can help with the challenging problem of generalizing to new elements.

**Force Norm Distribution Shifts.** Both MACE-OFF and GemNet-T deteriorate in performance when encountering systems with force norms different from those seen during training, as shown in Fig. 5. Interestingly, this performance drop occurs for both higher and *lower* force norms than those in the training set. Test-time training reduces errors for GemNet-T on out-of-distribution force norms, and also helps decrease errors for the new systems that are closer to the training distribution. The results in Fig. 5 specifically filter out new elements and different connectivity to isolate the effect of force norm distribution shifts.

**Connectivity Distribution Shifts.** For both MACE-OFF and GemNet-T, force errors increase when a test graph has a different connectivity than the training graphs. As a heuristic to describe connectivity, we calculate the percentage of Laplacian eigenvalues that are in $[0.9, 1.1]$, which is consistently around 55% for the training graphs in SPICE. Our test-time radius refinement (RR) technique (see §3.1) applied to MACE-OFF effectively mitigates connectivity errors at minimal computational cost. Test-time training also effectively mitigates connectivity distribution shifts, as shown in (Fig. 6 and Tab. 3). Note that Fig. 6 isolates connectivity distribution shifts by filtering out new elements and out-of-distribution force norms. See §C.3 for RR results with the JMP model on the ANI-1x dataset.

**Elemental Distribution Shifts.** Unsurprisingly, MACE-OFF and GemNet-T perform poorly when they encounter new elements at test time. Fig. 7 shows that test-time training can reduce errors on systems with new elements, sometimes by a factor of 10 for specific molecules.

While this is a challenging generalization task, we argue that achieving this should be a goal for a true chemistry foundation model, akin to first-principles methods that model the entire periodic table. Collecting more data for new elements is an option but can be prohibitively expensive, especially for atoms with many electrons. TTT provides a better starting point and reduces the amount of data that needs to be collected.

**Aggregated Results and Takeaways.** We present aggregated results on the SPICEv2 distribution shift benchmark, where a model is trained on SPICE and evaluated on 10k new molecules from SPICEv2. The large MACE-OFF foundation model trains on 951k samples but still suffers from distribution shifts on the new structures from SPICEv2. We also see that (1) the RR method mitigates connectivity distribution shifts for MACE-OFF at minimal computational cost (see Tab. 1) and (2) using TTT with the GemNet-T model performs better than MACE-OFF on the new molecules from SPICEv2, highlighting the effectiveness of training strategies for mitigating distribution shifts.

Since the improvements from RR and TTT are right-skewed, meaning many molecules show small improvements while some see large gains, we highlight the 10% of molecules with the greatest improvement in Fig. 5b, Fig. 6b, and Fig. 7b. We also present results for individual molecules in Tab. 2 and Tab. 3 to show that TTT and RR can help across a range of errors. Both TTT and RR improve results on molecules that already have low errors, and bring many molecules with high errors close to the in-distribution performance (see Fig. 10 which shows that more than $8{,}000/10{,}000$ molecules have errors below 25 meV / Å).

The ability of TTT and RR to mitigate distribution shifts supports the hypothesis that MLFFs easily overfit to training distributions, even with large datasets. By improving the connectivity and learning more general representations of test molecules, RR and TTT help diagnose the specific ways in which MLFFs overfit. These experiments suggest that improved training strategies could help learn more general models.

| | **Model** | | | |
|---|---|---|---|---|
| | MACE-OFF | GemNet-T | MACE-OFF + RR (ours) | GemNet-T + TTT (ours) |
| **SPICEv2 Test Set** (With New Elements) Force MAE (meV / Å) | 71.2±1.3 | 64.0±2.5 | 68.1±1.6 | **51.0±1.8** |
| **SPICEv2 Test Set** (Without New Elements) Force MAE (meV / Å) | 26.75±0.65 | 22.9±1.4 | 26.0±0.64 | **19.9±1.0** |

Table 1: **Aggregated Results on SPICEv2 Distribution Shift Benchmark.** We provide aggregated results on the SPICEv2 distribution shift benchmark with 95% confidence intervals. TTT and RR are both able to effectively mitigate errors across the 10k unseen molecules from SPICEv2.

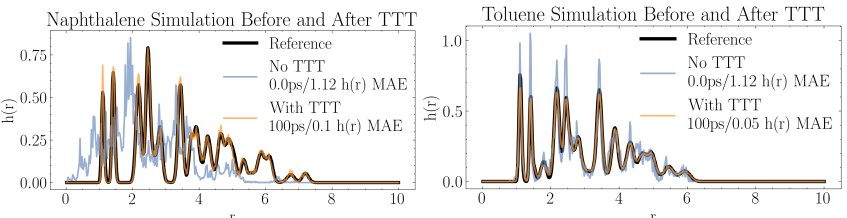

Figure 8: **TTT Enables Stable Simulations.** TTT enables stable simulations that accurately reconstruct observables, such as the distribution of interatomic distances, for molecules not seen during training (orange). In contrast, predictions without TTT for these unseen molecules result in unstable simulations and inaccurate $h(r)$ curves (blue). We also decreased the timestep by a factor of 5,000 and found that the simulations without TTT were still unstable.

## 4.2 EVALUATING GENERALIZATION WITH EXTREME DISTRIBUTION SHIFTS: SIMULATING UNSEEN MOLECULES

We establish an extreme distribution shift benchmark to evaluate the generalization ability of MLFFs on the MD17 dataset (Chmiela et al., 2017). We train a single GemNet-dT model (Gasteiger et al., 2021) on 10k samples each of aspirin, benzene, and uracil. We then evaluate whether this model can simulate two new molecules, naphthalane and toluene, which were unseen during training. Next, we evaluate whether TTT can address the distribution shifts to the new molecules. Using the same procedure outlined in §3.2, we pre-train on the 3 molecules in the training set with the sGDML prior, then freeze the representation model and fine-tune on the quantum mechanical labels. We then perform TTT before simulating the new molecules (see §3.2). This is an extremely challenging generalization task for MLFFs due to the limited variety of training molecules. Nevertheless, we believe that a model capable of accurately capturing the underlying quantum mechanical laws should be able to generalize to new molecules.

We evaluate the stability of simulations over time by measuring deviations in bond length, following Fu et al. (2023). We additionally calculate the distribution of interatomic distances $h(r)$ to measure the quality of the simulations. See §D for more details.

**Simulation Results.** As shown in Fig. 8, TTT enables stable simulations of unseen molecules that accurately reproduce the distribution of interatomic distances $h(r)$. Without TTT, the GemNet-dT model trained only on aspirin, benzene, and uracil is unable to stably simulate the new molecules and produces poor $h(r)$ curves. Even when we reduced the timestep by a factor of 5,000, the simulations without TTT remained unstable. We also found that TTT enables stable NVE simulations (see §C.2). Given that GemNet-dT + TTT can produce reasonable simulations without access to quantum mechanical labels of the new molecules, test-time refinement methods could be a promising direction for addressing distribution shifts.

## 5 CONCLUSION

We have demonstrated that state-of-the-art MLFFs, even when trained on large datasets, suffer from predictable performance degradation due to distribution shifts. By identifying shifts in element types, force norms, and connectivity, we have developed methods to diagnose the failure modes of MLFFs. Our test-time refinement methods represent initial steps in mitigating these distribution shifts, showing promising results in modeling and simulating systems outside of the training distribution. The success of these methods provides insights into how MLFFs overfit, suggesting that while MLFFs are expressive enough to model diverse chemical spaces, they are not being effectively trained to do so. This may indicate that training strategies, alongside data and architecture innovations, will be important in improving MLFFs. Additionally, we have established benchmarks for evaluating the generalization ability of the next generation of MLFFs.

**Reproducibility Statement.** We have described in detail our diagnostic and test-time refinement experiments throughout §2, §3, and §4. Details on computational resources used are provided in §G. Further experimental details (including hyperparameters), are discussed in §D. Details on the distribution shift benchmarks are also provided in §D. Our code will be publicly released on GitHub.

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

## A   RELATED WORK

**Distribution Shifts.**    There is a long line of literature studying distribution shifts in the machine learning community, which we briefly summarize here. Sugiyama et al. (2007) demonstrated how to perform importance weighted cross validation to perform model selection under distribution shifts. Methods have been proposed to measure and improve the robustness of models to distribution shifts in images (Taori et al., 2020; Zhao et al., 2022) and language (Zhang et al., 2019). Numerous methods have been proposed to tackle distribution shifts including, but not limited to, techniques based on meta learning (Jeong & Kim, 2020) and ensembles (Zhou et al., 2021).

Recent work has also begun identifying generalization challenges with MLFFs. Deng et al. (2024) find that MLFFs systematically underpredict energy surfaces, and that this underprediction can be ameliorated with a small number of fine-tuning steps on reference calculations. Our experiments compliment the initial findings of underestimation in their paper, and we also identify other types of distribution shifts, like connectivty and atomic feature shifts. Our proposed test-time refinement solutions are also able to mitigate distribution shifts *without* any reference data, and they provide insights into *why* MLFFs are unable to generalize.

**Multi-Fidelity Machine Learning Force Fields.**    Previous work has explored training MLFFs using multiple levels of theory. Jha et al. (2019), Gardner et al. (2024), and Shui et al. (2022) leveraged cheap or synthetic data to improve data efficiency and accuracy. Ramakrishnan et al. (2015) popularized the $\Delta$-learning approach (Bogojeski et al., 2020), where a model learns to predict the difference between some prior and the reference quantum mechanical targets. Multi-fidelity learning generalizes $\Delta$-learning by building a hierarchy of models that predict increasingly accurate levels of theory (Giselle Fernández-Godino, 2023; Vinod et al., 2023; Forrester et al., 2007; Heinen et al., 2024). Making predictions in the hierarchical multi-fidelity setting corresponds to evaluating a baseline fidelity level and then refining this prediction with models that provide corrections to more accurate levels of theory in the hierarchy.

Our work differs from these works in several ways. We focus on developing training strategies that address distribution shifts. In contrast to prior multi-fidelity works, we learn *representations* from multiple levels of theory using pre-training, fine-tuning, and joint-training objectives. Rather than fine-tuning all the model weights like in Jha et al. (2019), Gardner et al. (2024), and Shui et al. (2022), we explore freezing and regularization techniques that enable test-time training. Our new test-time objectives update the model's representations when faced with out-of-distribution examples, improving performance on out-of-distribution systems. Multi-fidelity approaches by themselves do not tackle the challenge of transferring to new, unseen systems at test-time. Nevertheless, combining our training strategies with other multi-fidelity approaches presents an interesting direction for future work.

**Test-Time Training.**    The test-time training (TTT) framework adapts predictive models to new test distributions by updating the model at test-time with a self-supervised objective Sun et al. (2020). Sun et al. (2020) demonstrated that forcing a model to use features learnt from a self-supervised objective during the main task allows the model to adapt to out-of-distribution examples by tuning the self-supervised objective. Follow up work showed the benefits of TTT across CV and NLP, exploring a range of self-supervised objectives (Gandelsman et al., 2022; Jang et al., 2023; Hardt & Sun, 2024).

## B   DETAILS ON TEST-TIME TRAINING (TTT)

We elaborate on the details of our proposed test-time training (TTT) approach.

**Model setup.**    Our model consists of the representation model, the main task head, and the prior task head, with parameters $\theta_R$, $\theta_M$, and $\theta_P$ respectively:

1. The representation model, $\theta_R$, is designed to extract features useful for both the main and prior task heads. These parameters can be trained on both the cheap data from the physical prior and the expensive reference calculations. After pre-training, the representation parameters can be further refined through fine-tuning and test-time training.

2. The main task head, $\theta_M$, predicts the energies and forces generated by DFT calculations. This head specifically uses the high-accuracy, expensive quantum mechanical labels produced by DFT for training.

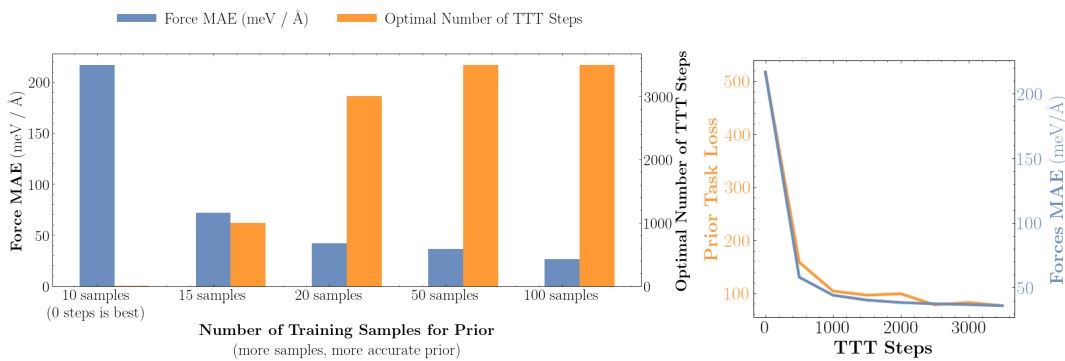

(a) **Prior Accuracy vs. TTT on Naphthalene.** As the prior becomes more accurate by training on more samples, we see larger improvements from TTT (blue bar). This accuracy allows us to take more gradient steps on the prior task (orange bar), without deteriorating performance on the main task.

(b) **Prior Task Loss vs. Main Task Loss.** Fitting to the prior task loss (orange) improves performance on the main task (blue) on Naphthalene.

Figure 9: **Understanding the Auxiliary Task in TTT.**

3. The prior head, $\theta_P$, predicts the energies and forces from the cheap physical prior, such as classical force fields. This head is trained with the cheap labels produced by the physical prior.

We emphasize that the pre-training and test-time training procedures described in §3.2 are model architecture agnostic. For details on how we split up existing architectures into the representation model, main task head, and prior head, see §D.

**Necessity of Proper Pre-training for Test-time Training.** The goal of TTT is to adapt to out-of-distribution test samples using a self-supervised objective at test-time (Sun et al., 2020; Jang et al., 2023; Gandelsman et al., 2022). In our case, we use the prior task loss $\mathcal{L}_P$ as the test-time training objective, making the model predict forces and energies labeled by the cheap physical prior. When an out-of-distribution (OOD) sample is encountered at test-time, we can adapt our representation parameters, $\theta_R$, using the prior. This update improves the features extracted from the OOD samples, which in turn smooths the potential energy surface and improves the performance on the main task (see Fig. 9b). Importantly, naive fine-tuning of the full pre-trained model (both $\theta_R$ and $\theta_M$) hinders the effectiveness of TTT. This is because fine-tuning $\theta_R$ on the main task may cause these parameters to "forget" the features learned from the prior during pre-training. If we adjust $\theta_R$ at test-time based solely on the prior targets, this could shift $\theta_R$ away from the representations that $\theta_M$ relies on to make predictions. Thus, for TTT to be successful, it is essential that the main task head depends on the features learned from the prior to make accurate predictions.

**Notes on the Prior.** Although the performance of TTT does improve with a more accurate prior (see Fig. 9a), we note that even in cases where the prior is poorly correlated with the main task (like with the EMT prior and OC20 in §C.5), TTT still provides benefits. This is because the prior is only used to learn *representations*, and *not* to directly make predictions on the targets. This means that as long as training on the prior yields good representations, it can be used for TTT.

We also argue that such a prior is in fact widely available. For instance, one could always train an sGDML prior on the existing reference data. Alternatively, one could use a simple potential (like EMT or Lennard-Jones). A different (cheaper) level of quantum mechanical theory can also be used. Alternatively, as with prior TTT work in computer vision, a fully self-supervised objective (like atomic type masking and reconstruction) could also be used. We leave explorations of more priors to future work.

It should be noted that using sGDML as the prior requires a few labeled examples to train the sGDML model for the unseen molecule. We show that as few as 15 labeled examples are sufficient to tune the prior and achieve good TTT results (see Fig. 9a). TTT also yields better results than fine-tuning directly on these 15 samples, since the model severely overfits on the small number of samples. We also emphasize that across the board, **TTT performs better than the prior** (see Tab. 14). In addition, the sGDML prior only works on one system, whereas the MLFF can model multiple systems.

**Limitations.** Test-time training incurs extra computational cost, mainly due to the gradient steps taken at test time. This cost is negligible compared to the overall training time of a model, and negligible compared to the time it takes to run simulation with the model. Additionally, our instantiation of TTT requires access to a prior. However, a suitable prior is almost always available since one can always use a widely applicable analytical or semi-empirical potential.

## C  ADDITIONAL TEST-TIME REFINEMENT RESULTS

We provide additional test-time refinement experiments using more models, datasets, and priors. Although these constitute challenging generalization tasks, test-time refinement shows promising first steps at mitigating distribution shifts and generalizing to new types of systems.

### C.1  FURTHER RESULTS ON SPICEv2 DISTRIBUTION SHIFT BENCHMARK

Since the TTT and RR results for the SPICEv2 distribution shift benchmark (see §4.1) are right skewed, there are many molecules that only improve slightly and a few that improve dramatically. In Tab. 2 and Tab. 3, we highlight results from 6 randomly selected molecules from the top 1,000 most improved with TTT and RR. Specifically, two molecules were randomly chosen from each of the following force error bins: $0–40, 40–100,$ and $> 100$ meV / Å. These results show that TTT and RR help across a range of errors: bringing high errors down to below 40 meV / Å, and improving results on already low errors.

| | $C_4NH_{12}N_3C_5H_3$ | $IC_2H$ | $ClOC_{14}NH_{15}C_{10}N_2C_3H_{14}$ | | | $O_3P$ |
|---|---|---|---|---|---|---|
| **GemNet-T** | 28 | 18 | 93 | 55 | 210 | 748 |
| Force MAE (meV/Å) / Stability (ps) | $100\pm0$ | $100\pm0$ | $14.7\pm1.2$ | $100\pm0$ | $100\pm0$ | $18.5\pm0.7$ |
| **GemNet-T + TTT** | 16 | 13 | 42 | 31 | 70 | 91 |
| Force MAE (meV/Å) / Stability (ps) | $100\pm0$ | $100\pm0$ | $38.2\pm6.0$ | $100\pm0$ | $100\pm0$ | $100\pm0$ |

Table 2: **Benefit of TTT.** We highlight specific examples from SPICEv2 where TTT provides large improvements. TTT can decrease errors by an order of magnitude, and can bring errors close to in-distribution performance. Even when errors are already low, TTT can further reduce errors. TTT also improves NVT simulation stability (mean $\pm$ standard deviation reported over 3 seeds).

| | $IC_2H$ | $O_5N_3C_{16}H_{35}$ | $N_4C_7H_{11}$ | $O_4C_2PH_6$ | $C_6N_2H_{12}$ | $SC_6H_4$ |
|---|---|---|---|---|---|---|
| **MACE-OFF** | 23 / | 12 / | 58 / | 79 / | 875 / | 109 / |
| Force MAE (meV/Å) / Stability (ps) | $100\pm0$ | $38.7\pm12.6$ | $100\pm0$ | $100\pm0$ | $62.8\pm26.3$ | $100\pm0$ |
| **MACE-OFF + RR** | 16 / | 9 / | 39 / | 49 / | 374 / | 69 / |
| Force MAE (meV/Å) / Stability (ps) | $100\pm0$ | $78.9\pm16.3$ | $100\pm0$ | $100\pm0$ | $100\pm0$ | $100\pm0$ |

Table 3: **Benefit of RR.** We highlight specific molecules from SPICEv2 to show that RR improves errors across a range of values. RR also improves NVT simulation stability (mean $\pm$ standard deviation reported over 3 seeds).

We also explicitly quantify in Fig. 10 that many molecular systems start with large errors and these errors are decreased to well within 40 meV / Å with TTT and RR. Additionally, hundreds of molecules across a range of errors have errors that are brought down significantly closer to the in-distribution performance. These results suggest that MLFFs are indeed expressive enough to model diverse chemical spaces, and can be better trained to do so.

**TTT is agnostic to the chosen prior.** Additionally, we explore using the semi-empirical GFN2-xT (Bannwarth et al., 2019) as the prior to provide further evidence that TTT is agnostic of the prior chosen. We train a GemNet-dT model with the pre-train, freeze, fine-tune approach described in §3.2 using GFN2-xT as the prior. The results in Tab. 4 show that TTT with GFN2-xTB also enables better performance across a range of errors.

| | | Overall | $O_2ClSNC_8$-$H_{16}$ | $O_2N_2C_{16}$-$SH_{14}$ | $O_3C_{19}$-$SiH_{26}$ | $O_2N_2C_{16}$-$SiH_{28}$ | $Cl_2C_7$-$SiH_{14}$ | $Cl_3C_9$-$SiH_{11}$ |
|---|---|---|---|---|---|---|---|---|
| **Force MAE** | **GemNet-dT** | 78.3±7.8 | 38 | 33 | 74 | 75 | 109 | 107 |
| **(meV / Å)** | **GemNet-dT + TTT** | 56.6±5.6 | 28 | 26 | 35 | 39 | 46 | 44 |

Table 4: **TTT on SPICE with a Semi-Empirical Prior.** We run TTT on our SPICEv2 distribution shift benchmark using the semi-empirical GFN2-xTB (Bannwarth et al., 2019) as the prior. TTT with a semi-empirical prior still improves results across a range of errors, bringing many molecules close to the in-distribution performance. 95% confidence intervals are reported for the overall error on the whole test set, and individual molecule examples are highlighted.

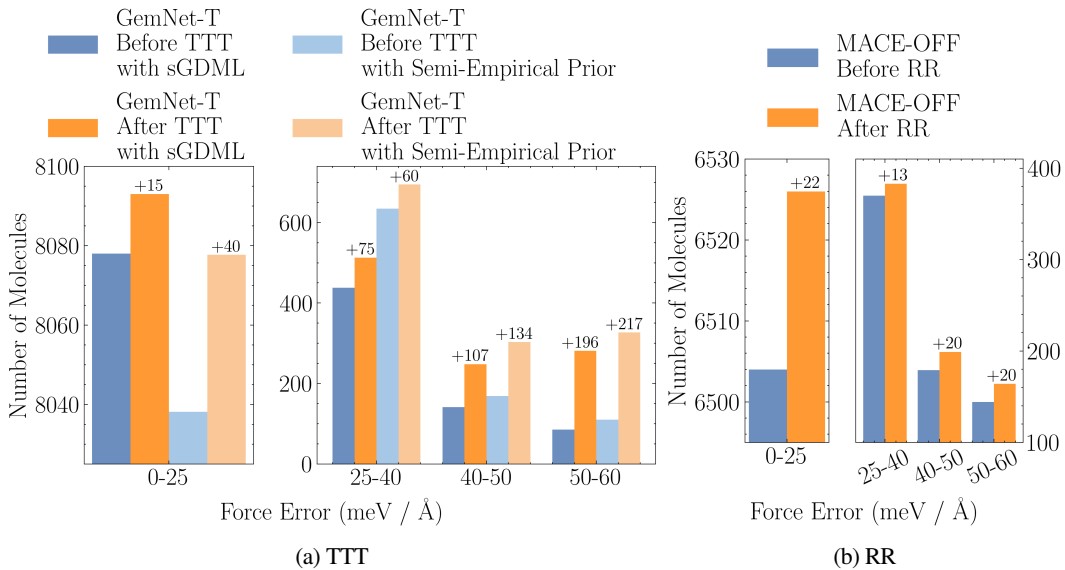

(a) TTT               (b) RR

Figure 10: **TTT and RR Help Accross a Range of Errors and Bring Many Molecules Close to the In-Distribution Performance on SPICEv2.** We plot the number of molecules that fall into specific force bins to show that TTT (a) and RR (b) help improve errors for hundreds of molecular systems. As with previous test-time training work, it is harder to improve performance the closer a system is to in-distribution (with lower errors).

| Molecule | GemNet-T | GemNet-T + TTT |
|---|---|---|
| Toluene | <1ps | 100 ± 0 ps |
| Naphthalene | <1ps | 43 ± 5.2 ps |

Table 5: **Stability of NVE Simulations with TTT.** TTT enables stable NVE simulations for molecules unseen during training on MD17. We report mean ± standard deviation across 3 seeds.

## C.2 NVE Simulations on MD17

We additionally run NVE simulations with the Velocity Verlet integrator (Hjorth Larsen et al., 2017) before and after TTT. As with the NVT simulations, we use a 0.5 fs time step and simulate for 100ps. Although simulations on naphthalene are slightly more unstable, TTT still increases the stability of simulations (see Tab. 5).

## C.3 Test-Time Radius Refinement with JMP on ANI-1x

We evaluate whether our proposed test-time radius refinement (RR) method (see 3.1) can help JMP (Shoghi et al., 2023) address connectivity distribution shifts in the ANI-1x dataset (Smith et al., 2020). Following

|  | Force Error Range (meV / Å) | | |
|---|---|---|---|
|  | 0-43 | 43-100 | >100 |
| **JMP on ANI-1x Test Set (Top 10%)** | 17.4±0.02 | 52.4±0.18 | 151.7±8.4 |
| Force MAE (meV/Å) | (15.1±0.07) | (52.3±0.54) | (167.7±39.3) |
| **JMP + RR (ours) on ANI-1x Test Set (Top 10%)** | 17.3±0.02 | 52.3±0.18 | 151.5±8.3 |
| Force MAE (meV/Å) | (14.6±0.07) | (51.9±0.54) | (163.6±37.8) |

Table 6: **Test-Time Radius Refinement with JMP on ANI-1x.** We implement our test-time radius refinement method (see §3.1) on JMP and evaluate improvements on the ANI-1x test set defined in Shoghi et al. (2023). Test-time radius refinement helps improve performance by mitigating connectivity distribution shifts. We highlight the top 10% of molecules with the greatest improvement in parentheses to show that test-time radius refinement helps across a range of errors.

| Example Molecules Force MAE Before → After RR (meV / Å) | | | | | |
|---|---|---|---|---|---|
| $C_3H_{10}N_2O_2$ | $C_5H_3NO$ | $C_5H_6N_2O$ | $C_5H_5NO_2$ | $C_5H_3N_3$ | $C_3H_6O_2$ |
| 6.9→5.4 | 8.2→6.2 | 53.0→44.2 | 85.2→78.3 | 101.1→99.7 | 158.9→149.7 |

Table 7: **Individual Examples from ANI-1x with RR on JMP.** We highlight individual molecular examples to show that RR helps across a range of errors.

the approach outlined in §4.1, we search over 7 different radius cutoffs from 6.5 to 9.5 Å to find the one that best matches the training Laplacian eigenvalue distribution.

As shown in Tab. 6 and Tab. 7, RR is able to improve force errors for JMP, including improving errors that are already low. We again highlight the top 10% of molecules with the greatest improvement, since the improvements from RR are right-skewed. RR often improves errors by 10-20% for individual molecules. This experiment provides further evidence that RR can address connectivity distribution shifts for existing pre-trained models at minimal computational cost, suggesting that existing models overfit to the graph structures seen during training.

### C.4 EVALUATING DISTRIBUTION SHIFTS IN THE MD22 DATASET: LOW TO HIGH FORCE NORMS

We establish a benchmark for force norm distribution shifts, using the MD22 dataset (Chmiela et al., 2023). The MD22 data set contains large organic molecules with samples generated by running constant-temperature (NVT) simulations, meaning that the majority of the structures are in lower energy states, and thus have low force norms. We filter out structures that have an average per-atom force norm smaller than a 1.7 eV / Å cutoff, which filters out about half of the data. We then evaluate whether GemNet-dT can generalize to high-force norm structures.

We train three different GemNet-dT models on 3 MD22 molecules—Ac-Ala3-NHMe, stachyose, and buckyball catcher—using the filtered low force norm dataset. We evaluate the GemNet-dT model on structures with force norms larger than the training cutoff. We also perform TTT using sGDML as the prior, as described in §3.2, to mitigate the distribution shift on the high-force norm test samples. For more details, see §D.

**Force Norm Generalization Results.** As shown in Tab. 8, GemNet-dT performs poorly on high force norm structures when compared to the low force norm structures it sees during training. TTT can mitigate the force norm distribution shift and close the gap between the in-distribution and out-of-distribution performance. This result further supports the hypothesis that MLFFs struggle to learn generalizable representations even when facing a distribution shift in a narrow single molecule dataset.

### C.5 TEST-TIME TRAINING ON OC20

The Open Catalyst 2020 (OC20) dataset consists of relaxation trajectories between adsorbates and surfaces (Chanussot et al., 2021). The primary training objective consists of mapping structures to their corresponding binding energy and forces (S2EF), as determined by DFT calculations. Both the S2EF task and OC20 dataset are challenging, due to the diversity in atom types and system sizes. The OC20 dataset includes an out-of-distribution test split consisting of systems that were not encountered during

| Force Norm Average | Model | Force MAE (meV / Å) | | |
|---|---|---|---|---|
| | | Ac-Ala3-NHMe | Stachyose | Buckyball Catcher |
| <1.7 eV / Å | GemNet-dT | 11.6 | 11.7 | 8.7 |
| >1.7 eV / Å | GemNet-dT | 36.8 | 24.2 | 16.4 |
| | ↓ | ↓ | ↓ | ↓ |
| | GemNet-dT + TTT | 26.5 | 19.0 | 12.7 |

Table 8: **Low to High Force Norms on MD22.** We train a GemNet-dT model on low force norm structures ($< 1.7$ eV / Å force norm averaged over atoms) and evaluate the model on high force norm structures ($> 1.7$ eV / Å). GemNet-dT generalizes poorly to the high force norm structures, but TTT significantly closes the gap.

Table 9: **OC20 test-time Training.** We evaluate a GemNet-OC model on the OC20 out-of-distribution validation split to assess the impact of joint-training and TTT.

| Model | Force MAE (meV/Å) | Energy MAE (meV) |
|---|---|---|
| GemNet-OC | 77.8 | 1787.4 |
| GemNet-OC Joint Training (ours) | 63.67 | 1320 |
| GemNet-OC Joint Training + TTT (ours) | **61.42** | **1143** |

Table 10: **TTT Hyperparameters for OC20 OOD Split.**

| Hyperparameter | Value |
|---|---|
| Steps | 11 |
| Learning Rate | 1e-4 |
| Optimizer | Adam |
| Weight Decay | 0.001 |

training. Even models trained on the full 100M+ OC20 dataset perform significantly worse on the out-of-distribution split (Chanussot et al., 2021). Consistent with previous test-time training work (Sun et al., 2020; Gandelsman et al., 2022; Jang et al., 2023), we use this split to assess our TTT approach.

**Problem Setup.** For our prior, we use the Effective Medium Theory (EMT) potential, introduced by Jacobsen et al. (1996). Using this, we can compute energies and forces for thousands of structures in under a second using only CPUs (Hjorth Larsen et al., 2017). The EMT potential currently only supports seven metals (Al, Cu, Ag, Au, Ni, Pd and Pt), as well as very weakly tuned parameters for H, C, N, and O. Consequently, we filter the 20 million split in the OC20 training dataset to only the systems with valid elements for EMT, leaving 600 thousand training examples. Similarly, the validation split is filtered and reduced to 21 thousand examples. While this work primarily focuses on evaluating our TTT approach, exploring the potential of a more general prior, or developing such a prior, represents a promising direction for future work.

**Training Procedure.** We use a joint training loss function, $\mathcal{L} = \mathcal{L}_P + \mathcal{L}_M$, to train a GemNet-OC model (Gasteiger et al., 2022), which is specifically optimized for the OC20 dataset. At test-time, we use the EMT potential to label all structures with forces and total energies. For each relaxation trajectory in the validation dataset, we update our representation parameters with the prior objective, $\mathcal{L}_P$ (see Eq. 6), and then make predictions with the updated parameters (see Eq. 7). The TTT updates are performed individually for each system in the validation set. See Tab. 10 for hyperparameters.

**Results.** We compare the performance of our joint-training plus TTT method against a baseline GemNet-OC model trained only on DFT targets and evaluated without TTT on the validation set. Despite the weak correlation between EMT labels and the more accurate DFT labels (see Fig. 11), using EMT labels for joint-training helps regularize the model and improves performance on the out-of-distribution split. After joint-training, implementing test-time training steps further improves the model's performance (see Tab. 9). This demonstrates that even though EMT has limited predictive accuracy as a prior, it can still be used to learn more effective *representations* that generalize to out-of-distribution examples. This experiment provides further evidence that improved training strategies can help existing models address distribution shifts.

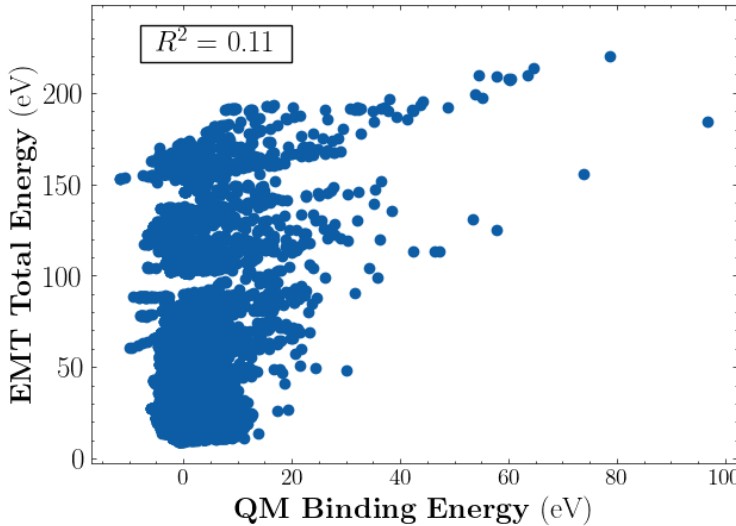

Figure 11: **EMT Correlation with Reference Energy DFT Calculations on OC20.** The correlation is very weak.

## D    EXPERIMENT DETAILS

We describe in detail the benchmarks established in this paper along with experiment hyperparameters. Code for benchmarks and training methods will be made available.

In line with previous test-time training works (Sun et al., 2020; Gandelsman et al., 2022; Jang et al., 2023), we update as few parameters as possible during TTT. For MD17, MD22, and SPICE experiments, we train everything before the second interaction layer in GemNet-T/dT. For OC20 (see §C.5), we train everything before the second output block in GemNet-OC.

Hyperparameters were largely adapted from Fu et al. (2023), although we increased the batch size to 32 to speed up training for GemNet-dT. Other deviations from Fu et al. (2023) are mentioned below.

### D.1    SPICEv2 DISTRIBUTION SHIFT BENCHMARK

**Dataset Details.**    We evaluate models trained on MACE-OFF's training split (Kovács et al., 2023), consisting of 951k structures primarily from the SPICE dataset (Eastman et al., 2023). The test set contains 10,000 new molecules from SPICEv2 (Eastman et al., 2024) not seen in the MACE-OFF training split. The 10,000 molecules were chosen to be the molecules that had the most structures in order to provide a large test set of 475,761 structures. GemNet-T was trained on the same data as MACE-OFF.

To evaluate the models on new elements, we found that replacing unknown elements with the closest known element from the periodic table to be simple and work well. We leave further investigation into representing new elements (such as interpolating between embeddings) to future work.

**Simulation Details.**    We run simulations for 100 ps using a temperature of 500K and a Langevin thermostat (with friction 0.01), otherwise following the parameters used in Fu et al. (2023). Since the SPICEv2 dataset was not generated purely from MD simulations, we do not have reference $h(r)$ curves for this dataset and instead focus on stability.

**Hyperparameters.**    Hyperparameters were adapted from Fu et al. (2023), with the following modifications shown to scale the model to 4M parameters to be more in line with MACE-OFF's 4.7M parameters:

1. Atom Embedding Size: $128 \rightarrow 256$

2. RBF Embedding Size: $16 \rightarrow 32$

3. Epochs: 250

| Parameter | Value |
|---|---|
| Learning Rate | 1e-4 |
| Momentum | 0.9 |
| Optimizer | SGD |
| Weight Decay | 0.001 |
| Steps | 250 |

Table 11: **TTT Parameters for SPICEv2 Distribution Shift Benchmark.**

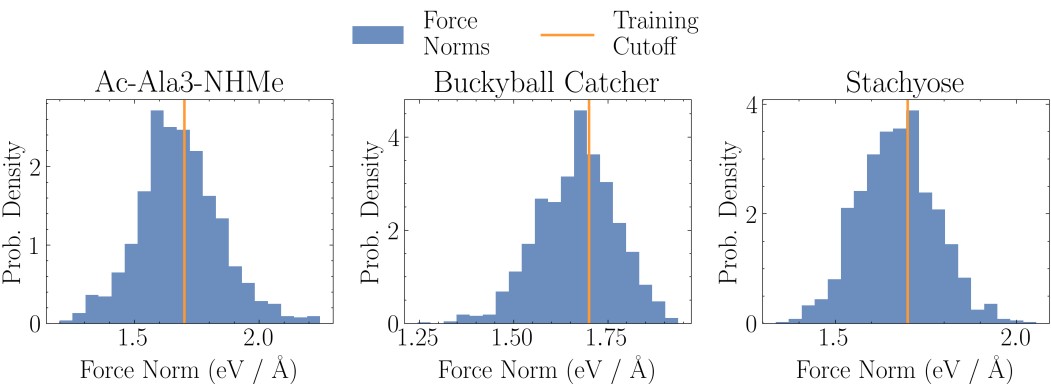

Figure 12: **Force Norms for MD22 Force Norm Distribution Shift Experiment.** Note that since the dataset was generated with NVT simulations, force norms are generally low when compared to SPICE.

Table 12: **TTT Hyperparameters MD22 Experiments.** We note that especially in cases where the prior is reasonably accurate, TTT is generally robust to a wide range of hyperparameter choices.

| Hyperparameter | Value |
|---|---|
| Steps | 50 |
| Learning Rate | 1e-5 |
| Optimizer | SGD |
| Momentum | 0.9 |
| Weight Decay | 0.001 |

For test-time training parameters, see Tab. 11. Note that we performed early stopping if the prior loss got stuck, or if it reached the in-distribution loss (since this implies overfitting and deteriorates performance on the main task).

### D.2    MD22 Low to High Force Norms Experiment

**Dataset Details.**    We train on approximately 6k samples from each molecule, corresponding to the 10% split for Ac-Ala3-NHME, 25% for stachyose, and 100% for buckyball catcher.

**Hyperparameters.**    See Tab. 12 for details on the hyperparameters used.

### D.3    Simulating Unseen Molecules on MD17

We provide further experimental details for the simulating unseen molecules benchmark on MD17 (see §4.2).

**Dataset Details.**    We use the 10k dataset split for the 3 training molecules (aspirin, benzene, and uracil). For test-time training, the 1k test-set is used for naphthalene and toluene. We note that TTT can also be done with structures generated from simulations with the prior, and we think further experimentation with this is an interesting direction for future work.

| Parameter | Value |
|---|---|
| Learning Rate | 1e-3 |
| Momentum | 0.9 |
| Optimizer | SGD |
| Weight Decay | 0.001 |
| Steps | 3000 |

Table 13: **TTT Parameters for MD17 Transferability Benchmark.**

| Molecule and Number of Training Samples (or source) | Force MAE (meV/Å) |
|---|---|
| **Naphthalene** | |
| 10 samples | 444.03 |
| 15 samples | 123.98 |
| 20 samples | 51.77 |
| 50 samples | 42.28 |
| 100 samples | 20.86 |
| **Toluene** | |
| 50 samples | 44.82 |
| **Ac-Ala3-NHMe** | |
| (Chmiela et al., 2023) | 34.25 |
| **Stachyose** | |
| (Chmiela et al., 2023) | 29.05 |
| **Buckyball Catcher** | |
| 100 samples | 99.15 |
| **Average over 10k molecules from SPICEv2** | |
| $\sim$20 samples | 62.25 (up to 724.5) |
| **EMT** | |
| (Jacobsen et al., 1996) | 415 |
| **GFN2-xTB on SPICEv2** | |
| (Bannwarth et al., 2019) | 201.6 |

Table 14: **Accuracy of Prior for TTT.** TTT always outperforms the prior.

**Simulation Details.** We run simulations for 100 ps using a temperature of 500K and a Langevin thermostat (with friction 0.01), otherwise following the parameters used in Fu et al. (2023). We measure the distribution of interatomic distances $h(r)$ to evaluate the quality of the simulations. The distribution of interatomic distances is defined as:

$$h(r) = \frac{1}{n(n-1)} \sum_{i}^{n} \sum_{j \neq i}^{n} \delta(r - \|\mathbf{x_i} - \mathbf{x_j}\|), \tag{8}$$

where $r$ is a reference distance, $\mathbf{x_i}$ denotes the position of atom $i$, $n$ is the total number of atoms, and $\delta$ is the Dirac Delta function. The MAE between a predicted $\hat{h}(r)$ and a reference $h(r)$ is given by:

$$\text{MAE}(\hat{h}(r), h(r)) = \int_{0}^{\infty} |\langle h(r) \rangle - \langle \hat{h}(r) \rangle| dr, \tag{9}$$

where $\langle \cdot \rangle$ indicates time averaging over the course of the simulation.

In both cases, TTT brings down force errors from $\sim$200 meV / Å down to less than 25 meV / Å, beating the prior (that uses 50 samples) and enabling stable simulation. We found that a prior that uses only 15 samples still leads to improvements with TTT (see Fig. 9a).

**Hyperparameters.** See Tab. 13 for hyperparameters used in the MD17 simulation experiments.

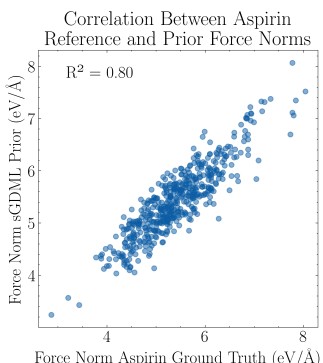

Figure 13: **Prior and Reference Force Norms are Highly Correlated.** We plot force norms calculated by the sGDML prior and the reference DFT for samples of aspirin from the MD17 dataset. The force norm predicted by the prior is highly correlated with the reference force norm, despite the absolute error between them being large.

## E  DETAILS ON DISTRIBUTION SHIFTS

We emphasize that element, force norm, and connectivity distribution shifts define "orthogonal" directions along which a shift can happen in the sense that they can each happen independently. In other words, a structure might have the same connectivity and similar force norms, but contain a new element. Similarly, for the SPICEv2 dataset, the distribution of connectivities is the same independent of force norm of the structure (see Fig. 14). This implies that one can observe a force norm shift while still seeing similar elements and connectivity.

Additionally, we provide more details on how we diagnose distribution shifts for new molecules at test time.

1. Identifying distribution shifts in the atomic features $\mathbf{z}$ is straightforward: one can simply compare the chemical formula of a new structure to the elements seen during training.

2. To diagnose force norm distribution shifts, we observe that although priors often have large absolute errors compared to reference calculations, *force norms* are actually highly correlated between priors and reference values (see Fig. 13 for an example from MD17). To determine whether a structure might be out-of-distribution with respect to force norms, the prior can be quickly evaluated at test time, and the predicted force norm can be compared to the training distribution.

3. Connectivity distribution shifts can be quickly identified by comparing graph Laplacian eigenvalue distributions. Based on our heuristic described in §3.1, we characterize graph connectivities by calculating the percentage of graph Laplacian eigenvalues that fall within the range of $[0.9, 1.1]$ for each molecule in a dataset. We then average this percentage across training molecules to represent the training distribution. We experimented with more involved versions of representing the training distribution, such as averaging probability density functions of eigenvalue distributions, but found that this made little difference since the connectivity was relatively uniform across the training molecules in SPICE, MD17, and MD22 (generally regular well-connected graphs). At test time, one can quickly calculate the percentage of eigenvalues that fall in $[0.9, 1.1]$ and compare this to the training distribution to identify connectivity distribution shifts. We think it is an interesting direction for future research to further study distribution shifts in graph structure.

We emphasize that our proposed methods for diagnosing distribution shifts are computationally efficient, and they do not require access to reference labels.

## F  THEORETICAL MOTIVATION FOR TEST-TIME RADIUS REFINEMENT

Our test-time radius refinement strategy is based on the theoretical finding presented by Bechler-Speicher et al. (2024), which states that GNNs tend to overfit to generally regular and well-connected training graphs. Although the theorems are presented for classification problems, they provide intuition and motivation for

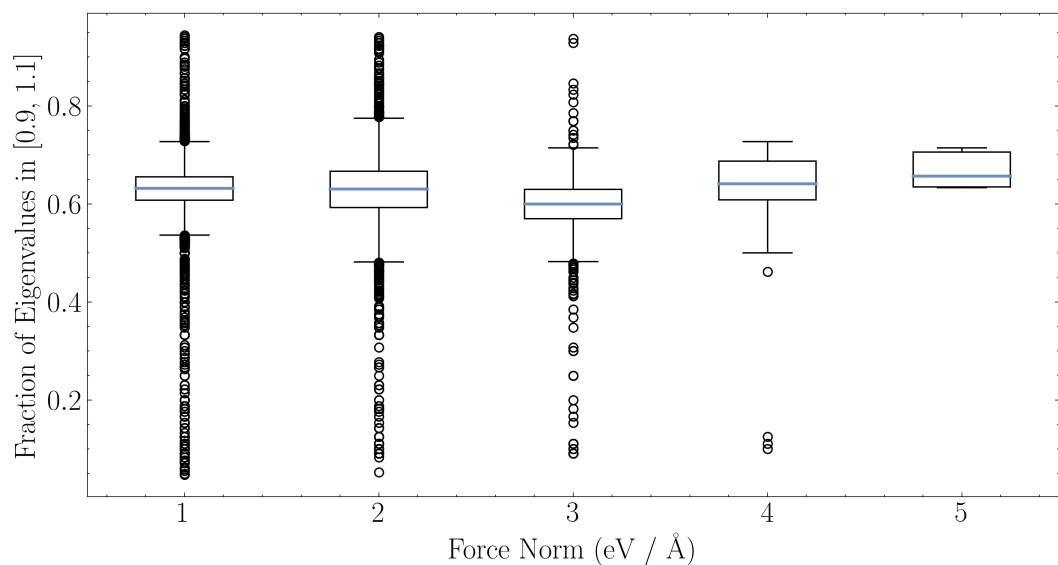

Figure 14: **Force Norm vs. Connectivity on SPICEv2.** The distribution of connectivities is similar across force different force norms. This implies that these distribution shifts can happen independently.

our RR approach. We restate some of the important theoretical results here (for the proofs and more details see Bechler-Speicher et al. (2024) and Gunasekar et al. (2019)).

**Theorem F.1 (Extrapolation to new graphs can fail (Bechler-Speicher et al., 2024))** *Let $f^*$ be a graph-less target function (it does not use a graph to calculate its output). In other words, $f^*(X,A) = f^*(X)$, where $X$ are node features and $A$ is the adjacency matrix of a graph. There exist graph distributions $P_1$ and $P_2$, with node features drawn from the same fixed distribution, such that when learning a linear GNN with gradient descent on infinite data drawn from $P_1$ and labeled with $f^*$, the test error on $P_2$ labeled with $f^*$ will be $\geq \frac{1}{4}$. In other words, the model fails to extrapolate to the new graph structures at test time.*

Mapping this to MLFFs, theorem F.1 suggests that a GNN trained on specific types of molecular structures (i.e., acyclic molecules) could fail to generalize to new connectivities at test time (i.e., a benzene ring).

**Theorem F.2 (Extrapolation within regular graph distributions (Bechler-Speicher et al., 2024))** *Let $D_G$ be a distribution over $r$-regular graphs and $D_X$ be a distribution over node features. A model trained on infinite samples from $D_G, D_X$ and labeled by a graph-less target function $f^*$ will have zero test error on samples drawn from $D_X, D_{G'}$ (and labeled by $f^*$), where $D_{G'}$ is a distribution over $r'$-regular graphs.*

In other words, generalizing across different types of regular graphs is easier for GNNs. Based on these theorems and our observation that many molecular datasets (MD17, MD22, SPICE) contain generally regular and well-connected graphs, we are motivated to find ways to make testing graphs look more like the training distribution (generally regular and well-connected) to help the models generalize. The observation that graphs for MLFFs are often generated by a radius cutoff led us to develop the RR method presented in §3.1. See Fig. 15, which empirically shows that RR makes graphs more regular and brings them closer to the distribution of training connectivities, aligning with our theoretical intuition. We think it is an interesting direction for future research to continue exploring the theoretical properties of graph structure distribution shifts.

## G COMPUTATIONAL USAGE

All of our experiments were run on a single A6000 GPU.

- MD17/22: Training for 100 epochs on a single molecule takes 2 GPU hours. Option 2 from Fig. 4a (pre-training, freezing, then fine-tuning) took 2 hours for pre-training and then 2 hours

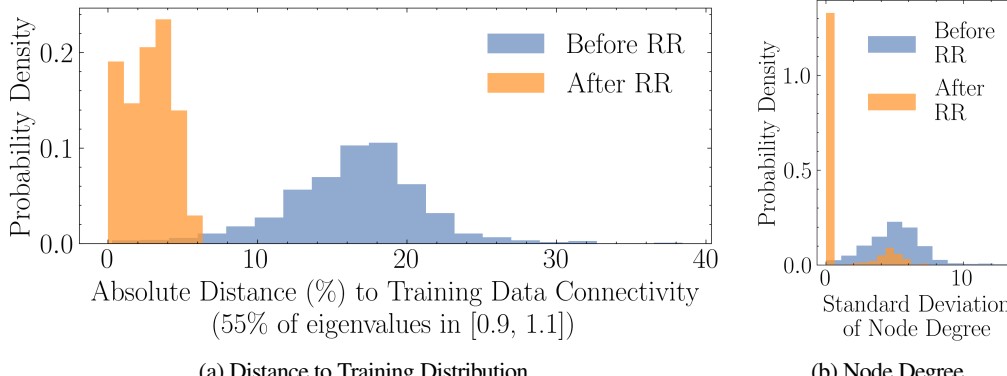

(a) Distance to Training Distribution            (b) Node Degree

Figure 15: **Radius Refinement (RR) Brings Connectivities Closer to the SPICE Training Distribution and Makes Graphs More Regular.** For those molecular systems where the connectivity does change by refining the radius (for some molecular systems the connectivity doesn't change unless the radius is made very small), we clearly see that RR decreases the difference in connectivity when compared to the training distribution (a). This is measured based on our eigenvalue heuristic. The SPICE training graphs had an average of 55% of their eigenvalues in the range of $[0.9, 1.1]$. We calculated the absolute difference between this training percentage, and the percentage of eigenvalues that fell in $[0.9, 1.1]$ before and after RR. After RR, the standard deviation of node degrees is smaller, indicating that the graphs are more regular (b).

    for fine-tuning (although we observed strong finetuning results with even less pre-training). TTT took less than 15 minutes for each molecule.

- SPICE Results: Pre-training on the prior took less than 5 hours on an A6000 across model sizes. Fine-tuning took 2 days. TTT took less than 5 minutes per molecule. In comparison, MACE-OFF small, medium, and large trained for 6, 10, and 14 A100 GPU-days respectively. Radius refinement takes less than 1 minute per molecule (to calculate eigenvalues to find the optimal radius).

- OC20: Joint-training (option 1) took 48 hours. Evaluation with TTT took 6 hours (compared to 2 hours without TTT).

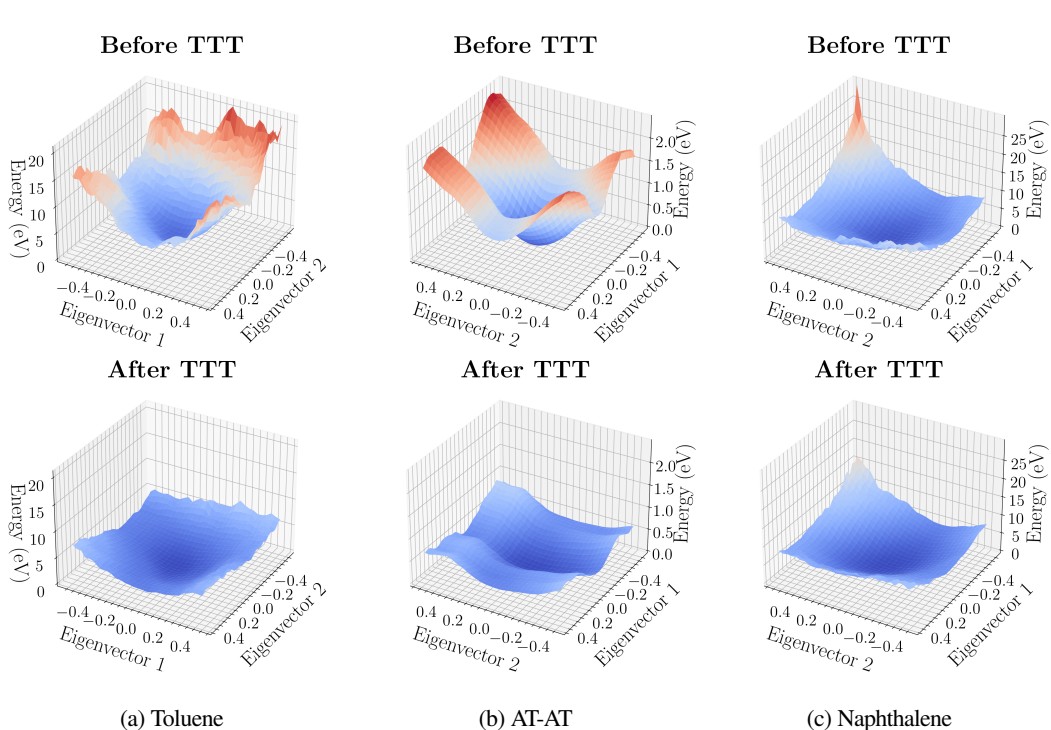

(a) Toluene

(b) AT-AT

(c) Naphthalene

Figure 16: **Predicted Potential Energy Surfaces for Molecules in MD17 and MD22.**

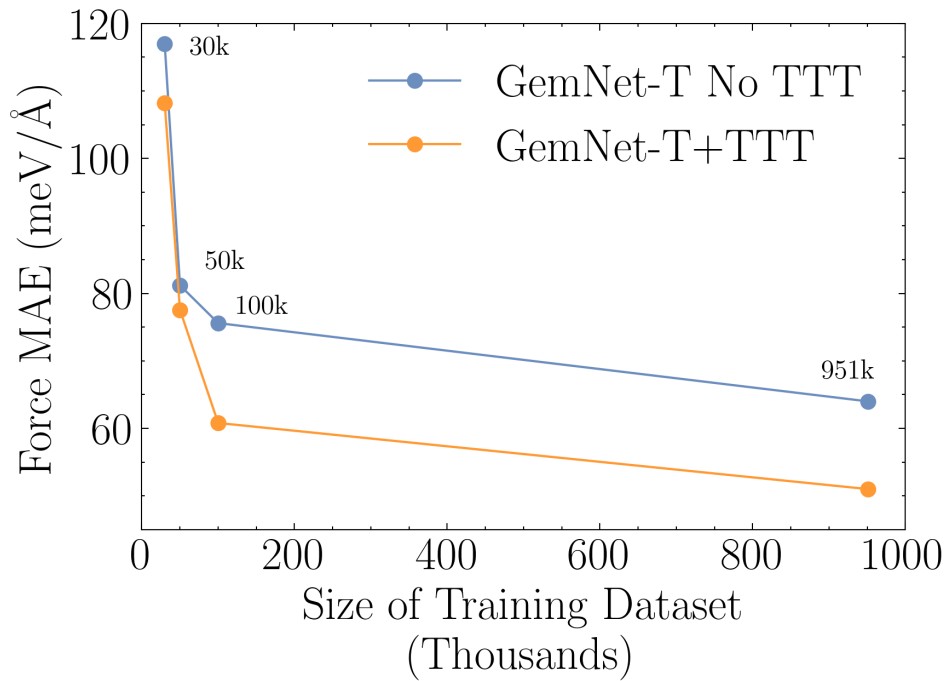

Figure 17: **Error (without TTT) on the SPICEv2 Distribution Shift Benchmark Versus Dataset Size.** Although increasing the amount of training data improves performance on the new SPICEv2 molecules, there are quickly diminishing returns. TTT helps across the board.

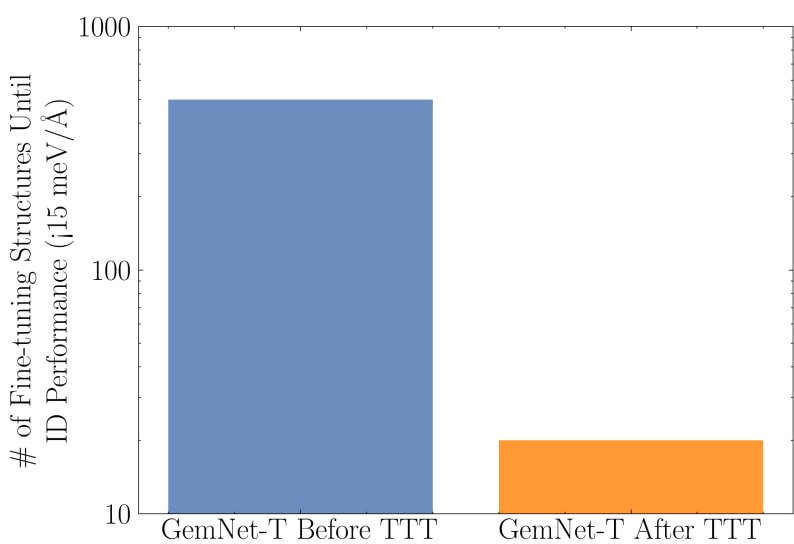

Figure 18: **TTT Enables Fine-tuning with 10x Less Data on MD17 to Reach In-Distribution Performance on Toluene.** We fine-tune a GemNet-dT model on a molecule not seen during training (toluene). Fine-tuning after TTT requires 10x less data to reach the in-distribution performance ($< 15$ meV / Å) compared to fine-tuning before TTT.

