# OpenReview forum: "Understanding and Mitigating Distribution Shifts for Machine Learning Force Fields"
_ICLR.cc/2025/Conference — Submitted to ICLR 2025_

### Official Review · Reviewer_UBE3 · 2024-10-25

**Soundness:** 3
**Presentation:** 3
**Contribution:** 3
**Rating:** 8
**Confidence:** 4

**Summary:**

This paper addresses the test-time domain-shift problem in machine learning force fields (MLFFs), a common issue when applying pretrained MLFFs to new materials in real-world applications. The authors identify three typical types of domain shifts and propose two test-time refinement strategies to address them. The authors provide partial experimental validation of their claims.

**Strengths:**

S1: The authors identify three primary origins of domain shift in MLFFs, most of which would not be explicitly appeared in the previous literature.

S2: They propose two strategies to mitigate these shifts: test-time radius refinement (RR) and test-time training (TTT) using inexpensive priors. As far as I know, this work would be the first one applying test time refinement strategies to MLFFs study.

S3: Experiments were conducted across various datasets, including MD17, MD22, SPICE, SPICEv2, and OC20.

**Weaknesses:**

Major Weaknesses:

W1. $\textbf{Assumption of Prior Knowledge on Test Data}$. The method assumes prior knowledge of test data (e.g., target materials, force labels). It conflicts with machine learning principles, where test data should ideally be inaccessible prior to testing, while this assumption may partially align with certain use cases in ab initio molecular dynamics simulations (MDs) studies (e.g., at least target material name and typical structure would be known beforehand). The scenario targeted in the paper should be more thoroughly explained in the abstract and introduction, including limitations and specific applications of the method.

W2. $\textbf{Lack of Clear Motivation}$. Related to the first point, the utility of the proposed method in practical settings is unclear. In computational chemistry, force MAE/RMSE values typically need to be below 1 kcal/mol/Å (~43 meV/Å) for reliability, but most results provided do not reach this threshold. Thus, practitioners may prefer simple data generation or active learning via DFT/CCSD for rigorously reliable results, leaving the proposed method of more academic than practical interest. This should be approached together with W1 in introduction, abstract, and conclusion (and maybe in limitation).

W3. $\textbf{Unclear Test Set Assumptions}$. Also connected to the first weakness, the assumptions regarding test set data (e.g., accessibility of force/potential labels, material structure details, availability of classical force fields) lack clarity. These setup should be explicitly defined in Section 2.

Minor Weaknesses:

W4. $\textbf{Potential Physical Limitations of Radius Refinement (RR)}$. Adjusting the cutoff radius in RR may introduce unexpected potential discontinuities, causing sudden and potentially destabilizing force changes (which is well-known issue for DFT/MLFFs). This modification can affect MD simulation results, and thus, a careful theoretical and empirical examination of the cutoff radius effect on MD simulations is advised.

W5. $\textbf{Insufficient MD Simulation Analysis.}$

(5.1) The authors conducted NVT simulations, which can artificially stabilize simulations due to the thermostat. A fair comparison between MLFFs with and without RR/TTT would benefit rather from NVE simulations.

(5.2) The MD simulation analysis includes only TTT, with no validation of RR effects. Based on point W4, it is strongly recommended to test RR's impact on MD simulations.

**Questions:**

Q1. How can the authors evaluate force distribution shifts without access to test dataset labels?

Q2. Why are there no experiments combining MACE+TTT and GemNet-T+RR?

Q3. Is TTT more accurate than models trained directly using prior labels, such as those from classical force fields?

Q4. Why were MACE, GemNet-T, and NequIP chosen as the MLFF models? Do they represent the broader MLFF landscape?

Q5. Why are "Necessity of Proper Pre-training for Test-Time Training" (Sec. C) and "Notes on the Prior" (Sec. F.4) in the Appendix? These seem critical and perhaps better to belong in the main text.

Q6. What does "the same element with a different charge" mean at line 150? If it refers to ionization, electrostatic forces should be considered, as this represents a significant problem setting change, not merely a distribution shift.

Q7. At lines 261-262, what does “for many molecular datasets” mean specifically? Additional information on the datasets would clarify this statement.

Q8. In Figure 1’s caption, “(middle)” appears to be missing after “high force norms (…).”

For additional feedback, refer to the "Weaknesses" section.

---

> ### Author Response · Authors · 2024-11-20
> **Response part 1 of 2**
>
> We would like to thank the reviewer for the thoughtful and constructive feedback. We are glad you found our identification and exploration of distribution shifts novel for the community.
>
> > **Comment:** The method assumes prior knowledge of test data
>
> We want to clarify that **we _do not_ assume knowledge of the test data prior to testing**. For instance, for the MD17 experiments, the model is first trained on prior labels from aspirin, benzene, and uracil, and then fine-tuned on the DFT labels for the same molecules. We do not assume anything about the testing molecules at this point. It is only at test time that the prior is used to quickly label the test molecules (naphthalene and toluene), taking less than 5 minutes (since the prior is very cheap). These labels are then used to conduct test-time training. For the radius refinement technique, again there is no assumption a priori about the test data: at test time the optimal radius is efficiently picked by comparing graph Laplacian eigenvalues. We have clarified our problem set up throughout the paper.
>
> > **Comment:** Most results provided do not reach chemical accuracy
>
> We note that for many individual molecular systems, the errors are brought down by up to an order of magnitude, and for many systems (more than 8,000 / 10,000) the errors are below 25 meV / Å on SPICEv2. In section 4.2 with MD17, errors are brought down from >200 to less than 22 meV / Å for both naphthalene and toluene, enabling stable simulations. Section C.4 has errors well below 30 meV / Å on MD22. **We have also added Fig. 10 that explicitly shows that many molecular systems start with higher errors and get errors brought down closer to ID performance with TTT and RR.**
>
> We note that our experiments **establish a lower bound on the performance gains that can only be improved with further architectural and training innovations.** Moreover, they help us understand how MLFFs generalize, and they also provide evidence for the hypothesis that MLFFs can be trained to generalize more effectively.
>
> > **Comment:** practitioners may prefer simple data generation or active learning
>
> We emphasize that our proposed methods can be combined with active learning / data generation and are not a replacement for them (see section 2.2 around line 210). By decreasing errors on OOD systems purely through improved training strategies, TTT and RR provide a better starting point for active learning / further fine-tuning, likely requiring less data collection. We leave further investigation combining TTT and RR with other methods for future work.
>
> > **Comment:** Clarification of problem setup and limitations in introduction, abstract, and conclusion (and maybe in limitation).
>
> We appreciate the feedback. We have added clarifications throughout the text on our problem setup. We reiterate important points here:
>
> - We do not assume access to test structures during training.
> - Labels from the prior can be quickly computed at test time, since the prior is very computationally efficient to compute (see appendix F).
> - We argue that a cheap prior is almost always widely available since one could always use a simple analytical potential (like EMT or Lennard-Jones). A different (cheaper) level of quantum mechanical theory or a semi-empirical potential (see appendix C.1 and Tab. 4 for an added experiment with GFN2-xTB) can also be used, which is available for many systems.
>
> See appendix B, E, and F for more details.
>
> > **Comment:** Adjusting the cutoff radius in RR may introduce unexpected potential discontinuities
>
> We note that RR does not introduce any new discontinuities compared to current approaches, since all MLFFs that use a radius cutoff or a k-nearest neighbor graph also experience discrete connectivity changes during simulations (since an atom can exit or enter the cutoff). We have also added clarifications on the theoretical motivation behind our method in appendix F.
> > **Comment:** Examination of the cutoff radius effect on MD simulations is advised.
>
> We have added MD simulations with models that use RR to Table 3, and find that RR is able to help improve MD stability.
>
> > **Comment:** NVE simulations
>
> We have run NVE simulations with our model on the MD17 dataset with results presented below and in Tab. 5 in appendix C.2 (reported mean plus minus standard deviation stability over 3 seeds).
>
>
> | Molecule    | GemNet-T | GemNet-T After TTT |
> |-------------|----------|----------------|
> | Toluene     | <1ps     | 100 +/- 0 ps     |
> | Naphthalene | <1ps     | 43 +/- 5.2  ps   |
>
> Even though GemNet-T + TTT is slightly less stable with NVE simulations on naphthalene compared to NVT, TTT still enables longer stable NVE simulations. We also emphasize NVT simulations are more commonly used [1], which is why we focused on those in the initial version.

---

> ### Author Response · Authors · 2024-11-20
> **Response part 2 of 2**
>
> > **Comment:** How can the authors evaluate force distribution shifts without access to test dataset labels?
>
> Since part of our paper is about understanding the distribution shifts that MLFFs face, we only used the test labels in Fig. 2 as a means of _understanding_ the failure modes of current state-of-the-art MLFFs. Similarly, in Fig. 5 the force norms were only used to generate the bins in the plot to help demonstrate the distribution shift. **We emphasize that no test labels were used at any point during the training / test-time refinement process.**
>
> Interestingly, we notice that despite the priors having large force MAEs compared to the ground truth, the force norms of the prior do correlate strongly with the reference calculations. Thus, the prior itself can be used to diagnose force norm distribution shifts. See the new Fig. 13.
>
> > **Comment:** Why are there no experiments combining MACE+TTT and GemNet-T+RR?
>
> We have added an experiment applying RR to the JMP foundation model, which uses a GemNet backbone (see Tab. 6 and 7, section C.3). Due to the computational cost of training MACE models on large datasets (>2 weeks) we focused on the GemNet model for the SPICEv2 experiment. MACE-OFF would have to be retrained using the pre-train with prior, freeze, finetune approach in order to use TTT.
>
> > **Comment:** Is TTT more accurate than models trained directly using prior labels, such as those from classical force fields?
> Yes, TTT is more accurate than the prior used, so it will also be more accurate than any model trained just on those labels. See Table 14 and the results in section 4.
>
> > **Comment:** Why were MACE, GemNet-T, and NequIP chosen as the MLFF models? Do they represent the broader MLFF landscape?
>
> Yes, they do represent the diversity of the broader MLFF landscape. We picked these because they are some of the best performing models on various modeling tasks, and they are widely used. They are the best on a number of different tasks [1][3].
>
> > **Comment:** Why are "Necessity of Proper Pre-training for Test-Time Training" (Sec. C) and "Notes on the Prior" (Sec. F.4) in the Appendix?
>
> Due to space limitations, we had put them in the appendix in the initial version. We adjusted and included more details about TTT in the main text.
>
> > **Comment:**  With ionization, electrostatic forces should be considered, as this represents a significant problem setting change, not merely a distribution shift.
>
> We agree, and we removed this. Charge is still a very hard problem, though we think it is an interesting direction for future work.
>
> > **Comment:**  At lines 261-262, what does “for many molecular datasets” mean specifically? Additional information on the datasets would clarify this statement.
>
> We have clarified in the text that we were referring to SPICE, MD17, MD22, and ANI-1x.
>
> > **Comment:**  In Figure 1’s caption, “(middle)” appears to be missing after “high force norms (…).”
>
> Thank you very much for the detailed feedback. We have added “middle” to the caption.
>
> [1] Xiang Fu, et. al. Forces are not enough: Benchmark and critical evaluation for machine learning force fields with molecular simulations. Transactions on Machine Learning Research, 2023.
>
> [2] Maya Bechler-Speicher, et. al. Graph neural networks use graphs when they shouldn’t, 2024. URL https://arxiv.org/abs/2309.04332.
>
> [3] Kovács, D., Batatia, I., Arany, E., & Csányi, G. (2023). Evaluation of the MACE force field architecture: From medicinal chemistry to materials science. The Journal of Chemical Physics, 159(4).

---

> ### Comment · Reviewer_UBE3 · 2024-11-21
> **reply**
>
> Thank you for your hard work to tackle the reviewers' comment.
>
> > It is only at test time that the prior is used to quickly label the test molecules (naphthalene and toluene),
>
> For me, this sentence shows that this method needs prior knowledge of the test data label (either pseudo label or true label). Can you please explain more why you can claim that you do not assume knowledge of the test data prior to testing? I have understood that test set prior knowledge is not used during training but TTT in theory require test data knowledge when utilized.
>
> > We note that our experiments establish a lower bound on the performance gains that can only be improved with further architectural and training innovations. Moreover, they help us understand how MLFFs generalize, and they also provide evidence for the hypothesis that MLFFs can be trained to generalize more effectively.
>
> > At a high level, we want to emphasize that our paper is about characterizing and better understanding distribution shifts in machine learning force fields. (General Remark)
>
> According to the modified draft (blue colored font part), I still cannot see the above claim enough in the abstract, introduction, and problem setting. Rather, now the claim is not consistent in the manuscript (partly saying the objective is to improve MLPI performance and other part saying the objective is more on the better understanding on distribution shift). The authors should clearly state the above claim in the draft more consistently.
>
> Overall, I think that the present draft still lacks consistency in terms of the motivation and objective as mentioned above, which would make the reader confused to evaluate this paper. In particular, I feel the definition of "test time" is unclear for me and should be clarified in the problem setting. For example, if it is purely ML terminology, TTT is cheating. However, if it means a real-world application case, TTT/RR can be interpreted as a good fine-tuning method ("test data" is the same training data but different label).
> I would really appreciate an effort to tackle this point for future readers.

---

> > ### Author Response · Authors · 2024-11-22
> >
> > > **Comment**: Can you please explain more why you can claim that you do not assume knowledge of the test data prior to testing?
> >
> > As the reviewer correctly notes, before testing, we do not assume anything about the test data. To clarify, at test time, we have access to the data, but we _do not_ have access to the ground truth labels (reference quantum mechanical calculations). This setup is the same as all other TTT works [1][2][3], where a self-supervised loss is used as an auxiliary objective at test time. For instance, in the case of TTT with masked autoencoding in computer vision [2], it is assumed to have access to images at test time but not the ground truth labels. When an image is encountered at test time, it is corrupted by masking out parts of the image. Gradient steps are then taken on a masked autoencoding loss, before making predictions on the label of interest (i.e. classification target).
> >
> > In our case, we also form an auxiliary objective with the test data at test time (noting that we do **not** have the ground truth labels). We take gradient steps using the prior as the target. These prior targets are quick to compute (thousands of iterations per second on a CPU), analogous to self-supervised learning TTT settings. Here, the prior targets take the role of the aforementioned autoencoder masked objective. The model does not have access to the ground truth label—classification target in the case of computer vision and quantum mechanical target in our case—at any point during testing.
> >
> > > **Comment**: The authors should clearly state the above claim in the draft more consistently.
> >
> > Thank you for the opportunity to clarify. We have made more changes (in blue) to the abstract, introduction, section 2 (around line 140), and section 3 (around 239) to clarify our motivation and objectives with the paper. Our paper is both about understanding distribution shifts and also about mitigating them through test-time refinement strategies. We characterize and understand the distribution shifts in section 2 in order to motivate the strategies presented in section 3. In turn, the mitigation strategies provide insights into why MLFFs are susceptible to distribution shifts, namely overfitting and poorly regularized representations (as opposed to only poor data or weak architectures).
> >
> > > **Comment:** The definition of "test time" is unclear for me and should be clarified in the problem setting.
> >
> > Although it is counterintuitive to update the model at test time, adapting models at test time is a recent problem set up explored extensively in computer vision (CV) and NLP [1][2][3]. **We emphasize that test-time training is not cheating since no test labels are used to update the model.** At test time, the model has access to a test sample (like an image in CV or a molecular structure in our case), and it is only with the test sample (and through an auxiliary loss, like a self-supervised objective) that the model is improved. We have clarified this throughout the paper, including defining test time in section 3.
> >
> > [1] Yu Sun, et. al. Test-time training with self-supervision for generalization under distribution shifts, 2020.
> >
> > [2] Yossi Gandelsman,et. al. Test-time training with masked autoencoders. Advances in Neural Information Processing Systems, 2022.
> >
> > [3] Minguk Jang, Sae-Young Chung, and Hye Won Chung. Test-time adaptation via self-training with nearest neighbor information. arXiv preprint arXiv:2207.10792, 2023.

---

> ### Comment · Reviewer_UBE3 · 2024-11-22
> **reply**
>
> Thank you for the authors' serous work to approach my request.
> I am now basically satisfied with the paper and raise my score.
>
> >We emphasize that test-time training is not cheating since no test labels are used to update the model.
>
> I think that test data set is usually used to evaluate generalization ability of the model and I imagine TTT would induce an overfitting to test data set. One interesting question would be if the model performance on the original validation data set is the same even after TTT (which is not a request but just a comment out of curiosity).

---

> > ### Author Response · Authors · 2024-11-23
> >
> > Thank you for your response! We appreciate all the helpful comments and suggestions from the reviewer, and for helping us make the paper stronger. We comment on your point here:
> >
> > > **Comment:** One interesting question would be if the model performance on the original validation data set is the same even after TTT.
> >
> > This is an interesting point. It is true that TTT fits to specific test points [1][2][3], and this is repeated for different test points. There has been work doing an “online” version of TTT, where the model accumulates the TTT updates between test points [1], although most of the follow-up TTT work [2][3] does not do this online version. In other words, TTT does “overfit” on the auxiliary task to specific test points, but TTT steps could always be recomputed for new data points (i.e. for the validation set). We do think online TTT / continual learning research is an interesting direction for future work.

---

### Official Review · Reviewer_bnuv · 2024-10-31

**Soundness:** 3
**Presentation:** 3
**Contribution:** 3
**Rating:** 6
**Confidence:** 4

**Summary:**

This paper aims to analyze and mitigate the out-of-distribution problems for machine learning force fields. This paper proposes pertaining and test-time-training (TTT) using data generated from classical physical priors instead of QM calculations. The proposed method is evaluated on the SPICE dataset and is shown to improve OOD performance, including experiments on MD simulations. the experiments also show an analysis of different types of "OOD" in the context of molecular simulations.

**Strengths:**

- The ML force field is a direction of significant interest to the AI for Science community and ICLR audience.
- The OOD challenge is an important one for ML FFs, as we expect to use ML FFs in extrapolation tasks such as relaxation and MD simulations.
- The proposed method improves model OOD performance on a variety of tasks.
- The analysis of different flavors of OOD-ness and how the proposed method helps is valuable to the community.

**Weaknesses:**

The authors include an MD benchmark on the effectiveness of TTT which is great. However, it would be more interesting to test out the SPICE-trained models on MD simulation as a generalization to unseen molecules will be a more suited task for a SPICE-trained model.

Further, it will be interesting to see how the OOD-ness of a test molecule impacts MD performance. The authors show a reduction in Force MAE with TTT and its correlation with the OOD-ness metrics such as force norm and graph Laplacian eigenvalues. How would TTT impact MD stability under these different types of ODD-ness?

**Questions:**

Why does the author train the GemNet-T model with only 10% of the data? Clearly, the impact of TTT is perhaps higher with less training data. It would be ideal if the authors could include the results of a GemNet-T trained on the same data as the MACE model.

---

> ### Author Response · Authors · 2024-11-20
>
> We would like to thank the reviewer for the thoughtful and constructive feedback. We are glad that you are also interested in the problem of OOD for MLFFs and that our analyses are interesting for the community.
>
> > **Comment:** SPICE-trained models and MD simulation
>
> We have added MD simulation results in Table 2 and 3 for models trained on SPICE. TTT and RR help SPICE-trained models simulate more stably compared to without these strategies.
>
> > **Comment:** How the OOD-ness of a test molecule impacts MD performance.
>
> As mentioned above, we have added some simulations with SPICE trained models showing that TTT also improves stability of the simulation. In section 4.2, we also showed that TTT was able to mitigate distribution shifts enough to enable stable MD simulations on MD17. Unfortunately, we were not able to run MD simulations across the whole test set present in tables 5-7 because it is prohibitively computationally expensive (10,000 molecules * 3-4 hours per molecule = >30,000 hours), but our results on subsets of the test set show that there is stability improvement with TTT and that distribution shifts lead to worse MD performance (for example on MD17).
>
> > **Comment:** Why does the author train the GemNet-T model with only 10% of the data?
>
> We had initially run a GemNet-T model on only 10% of the data because training on the full dataset is computationally expensive (> 2 weeks of training time). We also noticed in small scale experiments that scaling from 30k-100k data points did not change the OOD performance very much. This is also in line with Fig. 2 and section 2, which suggest that adding more in-distribution data does not help these models generalize as well. Nevertheless, we have started a new experiment with a GemNet-T model trained on the same data as MACE and have reported the results for the model trained so far in Tab. 1. We will update the final results after the model has fully converged for the final version of the paper.

---

> > ### Comment · Reviewer_bnuv · 2024-11-21
> >
> > Thank you for your reply. I have two followup questions:
> >
> > 1. how are the molecules in table 2/3 selected?
> > 2. "We also noticed in small scale experiments that scaling from 30k-100k data points did not change the OOD performance very much." Could you include the experimental results in the paper?

---

> ### Author Response · Authors · 2024-11-21
>
> Thank you for the questions.
>
> > **Comment:** How are the molecules in table 2/3 selected?
>
> To demonstrate that RR and TTT are effective across a range of errors, we randomly selected six molecules from the top 1,000 most improved with TTT and RR. Specifically, two molecules were randomly chosen from each of the following force error bins: 0–40, 40–100, and >100 meV / Å. We have clarified this in appendix C.1. Note that we also added a new Fig. 10, which shows that there are many more molecules that also see strong improvements from TTT and RR.
>
> > **Comment:** Could you include the scaling experimental results in the paper?
>
> Certainly, we have included those results in Fig. 17. We also present them in the following table:
> |                                    | **Training Dataset Size** |         |          |          |
> |------------------------------------|---------------------------|---------|----------|----------|
> |                                    | **30k**                   | **50k** | **100k** | **951k** |
> | **Force MAE on SPICEv2 without TTT (meV / Å)** | 117.7                     | 81.2    | 75.6     | 64.0     |
>
> Based on the additional computational cost of training on the full dataset, paired with the diminishing returns we observed from 50k to 100k, we had included the 100k results in the original version. We have also now updated the paper to include results with the model trained on all the 951k structures in the dataset (same as MACE-OFF).

---

> > ### Comment · Reviewer_bnuv · 2024-11-23
> >
> > Thanks for your reply. It would be great to include the performance across training dataset sizes with TTT.
> >
> > Overall I remain positive about the paper and would keep my rating of 6.

---

> > > ### Author Response · Authors · 2024-11-25
> > >
> > > Thank you for your comments and for helping make the paper stronger. We have updated Fig. 17 including the TTT results across dataset size as suggested.

---

### Official Review · Reviewer_2dvj · 2024-11-03

**Soundness:** 2
**Presentation:** 3
**Contribution:** 2
**Rating:** 5
**Confidence:** 5

**Summary:**

The paper studies the problem of training machine learning force fields. Specifically, the paper aims to investigate the generalization ability of current MLFFs. To address the generalization problem, the paper proposes two methods: 1) use radius refinement to identify the test graph structure most consistent with the training graph distribution; 2) use cheap priors to perform test-time-training on the MLFF. The paper shows that the proposed method can effectively improve the generalization ability of current MLFFs on unseen molecules.

**Strengths:**

1. The paper studies existing foundation MLFFs and investigates their generalization ability, which could be helpful for the community.

2. The paper proposes an interesting approach to search for the most similar graph structure by tuning the radius for the test unseen molecules.

**Weaknesses:**

1. Even though the paper shows some improvements on unseen molecules, the performance is still magnitudes away from the desirable chemical accuracy. In other words, the proposed method could hardly be useful in practice. As the downstream task could be much more expensive (e.g., wet lab experiments or subject studies) than the simulations, accuracy is still the top priority for MLFF.

2. The actual distance for radius refinement is quite simple, and may not capture more detailed information in the graph connectivity. Also, for the train eigenvalues, since we could have multiple molecules during training, do we aggregate their overall eigenvalue distribution? It seems to me a more reasonable assumption would be that as long as there is one training molecule graph that is similar to the test, the generalization performance could be better. Also, I think more case studies should be provided for this part. I.e., how does the radius change for certain test molecules change its connectivity and make the graph similar to training molecules?

3. The idea of test-time-training is not novel, and many previous works have utilized low-cost priors for MLFF training/fine-tuning.

4. Some assumptions about the generalization issue may not be practical. For example, if the test dataset has unseen elements or the test molecule has a specific sub-structure that has never been encountered, it is probably not ideal to use the pre-trained MLFF for such a task.

Minor:
1. The main paper should include at least a brief summary of the related work.

**Questions:**

Please address the weaknesses part.

---

> ### Author Response · Authors · 2024-11-20
> **Response part 1 out of 2**
>
> We would like to thank the reviewer for the thoughtful and constructive feedback. We are happy that you find our analysis of current MLFFs interesting for the community.
>
> > **Comment:** Performance is still magnitudes away from the desirable chemical accuracy.
>
> We want to emphasize that **errors are not “magnitudes away” from chemical accuracy**. For many individual molecular systems, the errors are brought down by up to an order of magnitude, and for many systems (more than 8,000 / 10,000) the errors are below 25 meV / Å on SPICEv2. In section 4.2 with MD17, errors are brought down from >200 to less than 22 meV / Å for both naphthalene and toluene, enabling stable simulations. Section C.4 has errors well below 30 meV / Å on MD22. **We have also added Fig. 10 that explicitly shows that many molecular systems start with higher errors and get errors brought down closer to ID performance with TTT and RR.**
>
> We note that our experiments **establish a lower bound on the performance gains that can only be improved with further architectural and training innovations.** They can also help us understand how MLFFs generalize, and they provide evidence for the hypothesis that MLFFs can be trained to generalize more effectively.
>
>
> > **Comment:** The actual distance for radius refinement is quite simple, and may not capture more detailed information in the graph connectivity.
>
> This is an interesting point made by the reviewer. While we experimented with using more complicated distance metrics for measuring discrepancies between graphs (such as a KL-divergence between eigenvalue distributions), we found that it made minimal difference in performance compared to the heuristic used in the paper. We have clarified this in appendix E and F, and we think it is an interesting direction for future research to further study distribution shifts in graph structure.
>
> > **Comment:** Also, for the train eigenvalues, since we could have multiple molecules during training, do we aggregate their overall eigenvalue distribution?
>
> Yes, that is correct! Based on our heuristic, we calculate the percentage of graph Laplacian eigenvalues that fall within the range of [0.9, 1.1] for each molecule in the SPICE training dataset. We then average this percentage across training molecules to represent the training distribution. As mentioned above, we experimented with more involved versions of representing the training distribution, such as averaging probability density functions of eigenvalue distributions, but found that this made little difference since the connectivity was so uniform across the training molecules (generally regular well connected graphs). We have clarified this in appendix E and F.
>
> > **Comment:** As long as there is one training molecule graph that is similar to the test, the generalization performance could be better.
>
> This is an interesting point. However, a single data point with a “new” connectivity is not enough for the model to generalize to these structures. This can be seen in Fig. 6 where the errors smoothly go up for connectivities that are more poorly represented in the training distribution (shown in gray).
>
> > **Comment:** More case studies should be provided for RR
>
> We have added Fig. 15, to further clarify the effect of RR on graph connectivity. For those molecular systems where the connectivity does change by refining the radius (for some molecular systems the connectivity doesn’t change unless the radius is made very small), we clearly see that RR decreases the difference in connectivity when compared to the training distribution. This is measured based on our eigenvalue heuristic. The training graphs had an average of 55% of their eigenvalues in the range of [0.9, 1.1]. We calculated the absolute difference between this training percentage, and the percentage of eigenvalues that fell in [0.9, 1.1] before and after RR. We also observe that the standard deviation of node degree decreases after RR, indicating that graphs become more regular. This improved connectivity translates into the improved errors seen in Fig. 6 and Table 1.

---

> ### Author Response · Authors · 2024-11-20
> **Response Part 2 of 2**
>
> > **Comment:** The idea of test-time-training is not novel, and many previous works have utilized low-cost priors for MLFF training/fine-tuning.
>
> We appreciate the feedback. To the best of our knowledge, prior work has not used TTT in the field of MLFFs, although as you mention, TTT has been explored in other areas of machine learning. We provided an in-depth discussion comparing our methods to other multi-fidelity work in the field in appendix A. To summarize here, in contrast to previous multi-fidelity works, our TTT and RR methods explicitly tackle the problem of generalization. We also explore freezing and joint training approaches that explicitly enable addressing distribution shifts. Our experiments help us understand how current MLFFs fail to generalize, and they suggest that MLFFs can be more effectively trained to handle distribution shifts. Our approaches can also be combined with other multi-fidelity work, such as Delta learning, to further boost performance.
>
> > **Comment:** Some assumptions about the generalization issue may not be practical
>
> While we agree that generalizing to new elements and substructures is challenging, we argue that this should be a goal to aim for with foundation MLFFs, especially given that first-principles methods are mostly able to tackle arbitrary substructures. Our experiments in section 2 are targeted at *understanding* the generalization gaps that a good MLFF should be able to capture, which can help guide the development of new MLFFs. Our experiments in section 4 take first steps at mitigating these distribution shifts formalized in section 2. We show through TTT and RR that MLFFs are expressive enough to generalize to new elements and substructures (see Fig. 7b where errors on new elements are well below 40 meV / Å for the top 10% of molecules and section 4.2 where TTT enables simulation of new molecules). These results can only be further improved with architecture, data, and training innovations. This is intentionally designed as a challenging evaluation, and we highlight it because the community is moving towards more general models that aim for better OOD performance.
>
> > **Comment:** The main paper should include at least a brief summary of the related work.
>
> Due to space limitations, we had included only a short paragraph in the introduction about related works (lines 86-89) in the main text, and we had a longer section about the rest of the related works in appendix A. We have cut down the manuscript and reordered it to include more discussion of the related work in the main text.

---

> > ### Comment · Reviewer_2dvj · 2024-11-26
> > **Reply**
> >
> > Thanks for answering my questions. Some of my concerns have been addressed. However, there are two remaining concerns, and I want to keep my score at this point.
> >
> > 1. As also mentioned by other reviewers, even though, for some cases, the performance can reach chemical accuracy, for others, there is still a gap, making the method not entirely practical.
> >
> > 2. I feel that the work's technical contribution is limited. The RR part has some novelty, but I think a more thorough analysis of both the objective and the optimization can be achieved. Even though the authors show evidence that more complicated strategies might not bring benefits, I don't see an intuitive explanation for that.
> >
> > Related to 1, as also mentioned by the other reviewer, if accuracy is of interest, the user may prefer fine-tuning or active learning over TTT. The authors reply that TTT can be used together with fine-tuning/active learning but leave the investigation to future work, and I don't feel convinced by the argument.
> >
> > Also, I want to add that the user must evaluate whether TTT works or achieves a certain level of accuracy on molecules with limited data. I am basically questioning whether the proposed TTT is suitable for the MLFF task: as accuracy is critical to MLFF, and downstream tasks are potentially very expensive, there should be a reasonable amount of data that can help verify the effectiveness of the MLFF (or some theoretical guarantees based on the size of the available data). If the data is available, then the user can just do fine-tuning. It's still related to 1 that there are some cases where the TTT-trained MLFF does not meet the chemical accuracy. In such cases, how should the user know and modify if they don't have test data to verify?

---

> ### Author Response · Authors · 2024-11-27
> **Reply Part 1 of 2**
>
> We appreciate the reviewer’s comments and detailed feedback. We respond to specific points below.
>
> > **Comment:**  There is still a gap in performance to ID, making the method not entirely practical
>
> Our paper is about both **understanding** and mitigating distribution shifts. We characterize and understand the distribution shifts in section 2 in order to motivate the strategies presented in section 3. In turn, the mitigation strategies provide insights into why MLFFs are susceptible to distribution shifts, namely overfitting and poorly regularized representations (as opposed to only poor data or weak architectures).
>
> In addition to these mitigation strategies helping us understand how MLFFs fail to generalize, our test-time refinement strategies take first steps at mitigating distribution shifts. We agree that there is still a gap in performance to ID, and we emphasize that **matching in-distribution performance without access to reference labels is in general an extremely challenging machine learning problem**. Achieving in-distribution performance in an out-of-distribution task is an extremely high bar **since we are updating fewer parameters and not using ground truth labels**. Nevertheless, our new Fig. 10, section 4.2 MD17 results, and section C.4 MD22 results show that the majority of errors are below 25 meV / Å (>8,000 / 10,000 for SPICEv2, and all the MD17 and MD22 results). Beyond enabling stable simulations (section 4.2), TTT also drastically reduces the amount of data needed for further fine-tuning (see further details below and the new Fig. 18), showing practical utility.
>
>
> > **Comment:** I feel that the work's technical contribution is limited.
>
> We understand that the assessment of technical contributions can be somewhat subjective. To provide further clarity, we highlight and clarify some of the key contributions made by the paper, which we believe demonstrate its significance and impact:
>
> - We formalize multiple common distribution shifts faced by MLFFs.
> - We show that multiple state-of-the-art models predictably struggle with these distribution shifts.
> - We show how to address connectivity distribution shifts with our test-time radius refinement algorithm.
> - We describe how to implement test-time training for MLFFs with a cheap prior
> - We demonstrate with extensive experiments that our test-time refinement strategies are effective at mitigating distribution shifts.
> - We demonstrate that these mitigation strategies also provide insights into why MLFFs are susceptible to distribution shifts
>
> The success of these methods and the predictability of the distribution shifts suggest that MLFFs are indeed overfitting to training distributions, and that they can be better trained to generalize. Our experiments also provide benchmarks for evaluating the generalization ability of MLFFs.
>
>
> > **Comment:** I don't see an intuitive explanation for “more complicated strategies might not bring benefits”
>
> We have added appendix F, which provides theoretical motivation for why GNNs overfit to training graph structures and generalize better with regular graphs. Fig. 15 shows that empirically, RR brings graphs closer to the training distribution at test-time by making graphs more regular. Even though it is hard to formalize an “intuition” for this, based on the theoretical results discussed in appendix F, any method that makes graphs more regular would likely help mitigate connectivity distribution shifts. Although it is possible that more complicated methods for quantifying and modifying graph connectivity could bring benefits for certain systems, we did not see this clearly in empirical results and our current instantiation of RR still provides evidence that MLFFs do overfit to graph distributions.

---

> ### Author Response · Authors · 2024-11-27
> **Reply Part 2 of 2**
>
> > **Comment:** The authors reply that TTT can be used together with fine-tuning/active learning but leave the investigation to future work
>
> TTT can be readily used together with other methods. To illustrate this point, we have added Fig. 18 in the paper to show that **a model after TTT requires 10x less data when fine-tuning on an OOD molecule to get to the in-distribution performance (<15 meV / Å on MD17), compared to a model without TTT**. The results are shown in the following table:
>
> |                      | Number of Fine-tuning Samples Required to Reach ID Performance (<15 meV / Å) |
> |----------------------|:----------------------------------------------------------------------------:|
> | GemNet-dT Before TTT |                                     500                                     |
> |  GemNet-dT After TTT |                                      20                                      |
>
> > **Comment:** Some theoretical guarantees based on the size of the available data
>
> In general, guarantees about errors as a function of data are challenging to make precise in the field of deep learning. Although there are rigorous guarantees for simplified cases, like linear models [5][6], results for deep learning models often take the form of empirical scaling laws driven by large amounts of computing power [2][3]. In this regard, in the context of our paper, see Fig. 17 for a scaling experiment with TTT.
>
> This same issue holds for any fine-tuning or active learning method, where there are few theoretical guarantees to dictate how much data is needed to reach a certain level of performance. We do think further theoretical work on understanding fine-tuning in the field of deep learning is an interesting direction for research, and is likely a paper in and of itself [1][2][3][4].
>
> > **Comment:** How should the user know and modify if they don't have test data to verify?
>
> We note that this issue of evaluating a model with limited data also applies to any other active learning or fine-tuning techniques (or any machine learning method for that matter): if one wants to verify an answer on a test set, one needs to calculate the ground truth label. One option to evaluate MLFFs with limited data is to look at simulation stability to determine the accuracy of a model. However, we expect that this approach would only rule out very bad models. Additionally, when using TTT, the accuracy of the prior head can be used to estimate how close the main task error is to the in-distribution performance (see Fig. 9b).
>
> [1] Utkarsh Sharma, & Jared Kaplan. (2020). A Neural Scaling Law from the Dimension of the Data Manifold.
>
> [2] Jared Kaplan, Sam McCandlish, Tom Henighan, Tom B. Brown, Benjamin Chess, Rewon Child, Scott Gray, Alec Radford, Jeffrey Wu, & Dario Amodei. (2020). Scaling Laws for Neural Language Models.
>
> [3] Jordan Hoffmann, Sebastian Borgeaud, Arthur Mensch, Elena Buchatskaya, Trevor Cai, Eliza Rutherford, Diego de Las Casas, Lisa Anne Hendricks, Johannes Welbl, Aidan Clark, Tom Hennigan, Eric Noland, Katie Millican, George van den Driessche, Bogdan Damoc, Aurelia Guy, Simon Osindero, Karen Simonyan, Erich Elsen, Jack W. Rae, Oriol Vinyals, & Laurent Sifre. (2022). Training Compute-Optimal Large Language Models.
>
> [4] Roberts, D., Yaida, S., & Hanin, B. (2022). The Principles of Deep Learning Theory: An Effective Theory Approach to Understanding Neural Networks. Cambridge University Press.
>
> [5] Daniel Hsu, Sham M. Kakade, & Tong Zhang. (2014). Random design analysis of ridge regression.
>
> [6] Ng, A., & Jordan, M. (2001). On Discriminative vs. Generative Classifiers: A comparison of logistic regression and naive Bayes. In Advances in Neural Information Processing Systems. MIT Press.

---

### Official Review · Reviewer_xYhQ · 2024-11-03

**Soundness:** 3
**Presentation:** 2
**Contribution:** 2
**Rating:** 6
**Confidence:** 3

**Summary:**

The paper proposes two low-cost test-time refinement strategies to address distribution shifts in Machine Learning Force Field (MLFF) models, a significant challenge even for large foundation models trained on extensive datasets. Specifically, the first strategy leverages spectral graph theory to adjust the edges of test-time graphs, aligning them with structures observed during training, while the second strategy adapts representations for out-of-distribution (OOD) systems at test-time through gradient steps based on an auxiliary objective. The authors provide empirical results on several OOD benchmarks, demonstrating the effectiveness of these approaches.

**Strengths:**

* Given that expanding MLFF applicability to diverse chemical spaces is a primary goal within AI-for-Science, this approach is well-motivated and offers a promising step towards this objective.
* The paper establishes practical criteria for identifying distribution shifts, which may inspire future work, and the proposed refinement strategies demonstrate meaningful performance improvements over large-scale foundation models and pre-trained baselines on established benchmarks.
* The paper is well-organized and easy to follow.

**Weaknesses:**

* The proposed strategies yield only modest performance gains over pre-trained models, falling short of significantly improving OOD sample prediction accuracy to levels comparable with in-distribution (ID) samples or chemical accuracy. This limitation may hinder the practical applicability of the methods and affect the perceived technical contribution.
* The prior model used in the test-time training strategy (sGDML) may also face generalization issues on OOD samples, potentially limiting the effectiveness of pseudo-labels derived for fine-tuning. Did the authors consider using semi-empirical electronic structure methods (e.g., DFTB) as a prior model? Semi-empirical methods might offer better generalization in broader chemical spaces.

**Questions:**

* The idea of using the Laplacian spectrum to characterize and align graph structures for OOD and ID samples is interesting but would benefit from more theoretical or intuitive insights. Could the authors clarify the motivation behind aligning spectra as a strategy for managing distribution shifts?
* The accuracy differences reported in Table 1 (e.g., 26.75 vs. 26.0) appear slight. Could the authors provide statistical analyses to ensure these differences are not due to random fluctuations in test data?

**Details Of Ethics Concerns:**

No ethics concerns.

---

> ### Author Response · Authors · 2024-11-20
>
> We would like to thank the reviewer for the thoughtful and constructive feedback. We are glad that you think our investigation of distribution shifts is useful to the MLFF community.
>
> > **Comment:** falling short of significantly improving OOD sample prediction accuracy to levels comparable with in-distribution (ID) samples or chemical accuracy
>
> Matching in-distribution performance is a challenging open machine learning problem. Previous test-time training works are also unable to fully match ID performance [1][2]. Although we agree this is the ideal to reach, it is an unreasonably hard initial expectation, **since we are updating fewer parameters and not using ground truth labels**. Our initial results establish a lower bound on the performance gains that can only be improved with further architectural, data, and training innovations. One of our goals is to also help us understand how MLFFs generalize, and these initial training strategies provide evidence for the hypothesis that MLFFs can be trained to generalize more effectively.
>
> We do note that for many individual molecular systems, the errors are brought down by up to an order of magnitude, and for many systems (more than 8,000 / 10,000) the errors are below 25 meV / Å on SPICEv2.  In section 4.2 with MD17, errors are brought down from >200 to less than 22 meV / Å for both naphthalene and toluene, enabling stable simulations. Section C.4 has errors below 30 meV / Å on MD22. **We have also added Fig. 10 that explicitly shows that many molecular systems start with higher errors and get errors brought down closer to ID performance with TTT and RR.**
>
> >  **Comment:** Prior model used in the test-time training strategy (sGDML) may also face generalization issues.
>
> While it is true that the prior can also face generalization issues, we emphasize that the prior is only used to learn representations. Empirically we find that even in cases when the prior also generalizes quite poorly and has large errors (see for example Fig. 11), it can still be used to improve representations. In other words, as long as the prior can regularize the MLFF, it can still be used to improve test-time performance.
>
> > **Comment:** Using semi-empirical methods as the prior model
>
> This is a good point about trying a different prior.  Since our proposed **TTT method is agnostic of the chosen prior**, we have added an experiment to the appendix showing that a semi-empirical method (GFN2-xTB) can also be used with TTT (see appendix C.1 and Tab. 4).
>
> > **Comment:** Clarification of the motivation behind aligning spectra as a strategy for managing distribution shifts.
>
> Thank you for the comment. We have added further details to appendix F explaining some of the theoretical motivation for our RR approach. To summarize, [3] showed that GNNs struggle to generalize to new types of graph structures at test time. In particular, GNNs struggle more severely when graphs are not regular. Therefore, we were motivated to make the molecular graph structures at test time more similar to the training distribution (and more regular). To characterize graph connectivities, we study the eigenvalue distributions of Laplacian matrices. We find that by aligning spectra, we can make test graph connectivities that resemble training connectivities, making test connectivities more regular and helping MLFFs generalize better (see our new Fig 15 in the paper).
>
> > **Comment:** Statistical analyses for Table 1.
>
> We have added 95% confidence intervals to Table 1. As you mentioned, when considering systems from SPICEv2 without new elements, the improvements from RR are not significant in aggregate. All other differences in the table are significant (including TTT results and RR across the whole test set). We also note that the reason the differences are small in this table is because this is aggregated over the 10,000 test molecules from SPICEv2. Many molecules only improve a little with TTT and RR when the system is already very close to in-distribution. For individual molecules, the differences are much larger for both RR and TTT (see Tab. 2 and Tab 3). Our new Fig. 10 also highlights that many molecules improve significantly with RR and TTT.
>
> [1] Yu Sun, et. al. Test-time training with self-supervision for generalization under distribution shifts, 2020.
>
> [2] Yossi Gandelsman,et. al. Test-time training with masked autoencoders. Advances in Neural Information Processing Systems, 2022.
>
> [3] Maya Bechler-Speicher, et. al. Graph neural networks use graphs when they shouldn’t, 2024. URL https://arxiv.org/abs/2309.04332.

---

### Author Response · Authors · 2024-11-20

We appreciate the detailed and thoughtful feedback from all the reviewers. We summarize important updates and comments about the paper here. We have uploaded a new manuscript and updates to the manuscript are in blue. We also reply individually to reviewers with further updates and clarifications.

**At a high level, we want to emphasize that our paper is about characterizing and better understanding distribution shifts in machine learning force fields**. We also present some initial strategies to mitigate these shifts that **establish a lower bound on the performance gains that can only be improved with further architectural, data, and training innovations.** These topics are especially timely given that more large “foundation models” are being developed in this field, and we think that it is important to learn and characterize where such models are not doing well, in order to make them better. One of our goals is to also help us understand how MLFFs generalize, and these initial training strategies provide evidence for the hypothesis that MLFFs can be trained to generalize more effectively.

**We want to clarify that for out-of-distribution tasks (OOD), matching in-distribution performance (ID) is challenging for machine learning models [1][2].** While it is difficult in machine learning in general to exactly match ID performance on OOD tasks, our experiments on SPICE (section 4.1), MD17 (section 4.2), and MD22 (section C.4) show that TTT and RR are able to mitigate distribution shifts, and the majority of errors on the molecules are below 25 meV / Å (more than 8,000 / 10,000 for SPICE, and all the MD17 and section C.4 MD22 results). **We also have added Fig. 10 (to supplement Tables 2 and 3) to the paper to explicitly quantify that many molecular systems start with errors above 40 meV / Å and get errors brought down close to ID performance with TTT and RR.**




**We have included results with a GemNet-T model trained on 100% of the SPICE data (as suggested by reviewer bnuv).** We show that the same patterns hold in terms of increased errors with distribution shifts, helping us understand how MLFFs fail to generalize. TTT still significantly helps with OOD performance, and all the absolute errors are lower, providing further evidence that our results can only be improved with further data, training, and architecture innovations.

**We have added more molecular dynamics simulation experiments.** We show that TTT enables stable simulations with an NVE integrator. We also run MD simulations with models trained on the SPICE dataset, as well as with models that use our radius refinement technique, showing that our methods improve stability of the simulations. These can be found in Tab. 2, Tab. 3, and section C.2.

**We have added additional foundation models (EquiformerV2 and JMP) to section 2** to further help identify distribution shifts that foundation models face. We have also shown that test-time radius refinement (RR) works on the JMP model (which is built off of a GemNet backbone). See section 2 and Tab. 6 and 7 in the appendix.

**We have included further motivation and clarifying details about our RR approach.** To summarize, we were inspired by the theoretical analyses presented in [3] which show that GNNs easily overfit to regularly connected graphs. Our RR approach adjusts graph connectivity at test time to match the connectivity of graphs seen during training. We do so by aligning graph Laplacian eigenvalue distributions using the heuristic described in the paper. Although we explored more involved distance measures for aligning connectivities, we found that they had negligible performance improvements compared to our heuristic. We have added these clarifications to the manuscript in appendix F.

We thank the reviewers again for the helpful feedback. We believe these additional experiments and clarifications address the major issues brought up by the reviewers.

[1] Yu Sun, et. al. Test-time training with self-supervision for generalization under distribution shifts, 2020.

[2] Yossi Gandelsman,et. al. Test-time training with masked autoencoders. Advances in Neural Information Processing Systems, 2022.

[3] Maya Bechler-Speicher, et. al. Graph neural networks use graphs when they shouldn’t, 2024. URL https://arxiv.org/abs/2309.04332.

---

### Meta-Review · Area_Chair_sDLe · 2024-12-19

**Metareview:**

The paper proposes an analysis on types of distribution shift for learning and using machine learning force field (MLFF) and two ways to bridge them. This is an important topic of practical consideration, and the methods are investigated over a variety of datasets. While most reviewers appreciate the contributions, there still remain some concerns and insufficiencies. While I also appreciate the effort on this topic, I still feel that the paper seems a bit heuristic and is expected to provide more serious justifications on some points.
* The characterizations of distributional shift seems a bit superficial. The first type is more like an instance/sample-level difference: they are just different inputs. The second one needs further justifications considering that typical GNNs may already be able to output forces with a larger norm when the input atoms are closer. The authors may need to consider the built-in inductive bias of a model/architecture when discussing generalization. The third one also seems superficial: if the difference in the input graph structure can be accounted as a distribution shift, then why won't any change in the input, e.g., a different 3D conformation, is considered?
* The paper seems to not sufficiently distinguish instance-level difference and distribution-level difference. I hope to see more analysis on the distribution shift across different datasets, while the paper seems to present instance-to-distribution difference at most (for the cutoff radius selection). Moreover, how do the authors ensure that it is not due to the in-distribution generalization gap that including more in-distribution data still cannot bridge the gap (Fig. 2)?
* For the first method, why is graph Laplacian sufficient to represent the data distribution? This is only a one-dimensional projection of the data distribution, and might lose distribution in other aspects. I read from the authors' rebuttal and would like to further ask, does the ineffectiveness of using KL-divergence between eigenvalue distributions indicate that the proposed method may not be working in the designed way?
* Also regarding the first method, why changing cutoff radius won't break the model's knowledge on inter-atomic interaction? More specifically, in training, the model may have aggregated effects of atoms outside the cutoff radius to those within the radius. Will the model double-count the effects when using a larger radius? Or if the model would miss necessary interactions when using a smaller radius?
* Pretraining by aligning with a classical force field is not essentially a novel one. It could be very similar to pretraining a model on a large number of cheaper labels. The authors may also be interested in discussing why this won't make the model inherit artifacts (e.g., overly flat energy landscape) of the classical force field, which would make a noticeable difference on off-equilibrium structures.
* Some reviewers also complained the final performance not being useful as there is still a gap to the in-distribution case. This is indeed an insufficiency, but I won't put too much criticism on this due the intrinsic distribution shift gap (as the authors argued). As the method does not require direct supervision on the test domain, an improvement could already be beneficial.

I have gone through reviewers' comments. Although more reviewers have proposed a score suggesting acceptance and I'm aware of the contributions, and one reviewer has raised the score, they still raised a few insufficiencies, with some still remain. Moreover, I did not find my concerns sufficiently addressed throughout the rebuttals. I hence recommend a reject and hope the authors could further improve the paper for a better version.

**Additional Comments On Reviewer Discussion:**

* All reviewers appreciated the importance of the problem and the practical improvements. Some reviewers also mentioned the contribution of the characterization of distribution shift, though I still feel it a bit superficial (my first bullet point).
* Reviewers xYhQ, 2dvj, and UBE3 complained the absolute performance is still not satisfying (e.g., not reaching chemical accuracy). In the rebuttal, the authors explained that the achieving the same performance in an out-of-distribution setting as in the in-distribution setting should not be expected, and the shown improvement is arguably in the same level as the in-distribution case. The advantage in a finetuning setting is also shown in the rebuttal. I did not find clear further response from the reviewers, but I generally agree with authors' point (my last bullet point).
* Reviewers bnuv and UBE3 asked about evaluation for MD simulation. The authors provided such results in the rebuttal, which seem to be satisfying.
* Reviewers bnuv and UBE3 asked about results that crosses the considered models and datasets. The authors provided further results of GemNet on SPICE. Though not directly the one asked due to computational cost, the provided results seem to convey the similar information. Evaluation results with various data set sizes are also appreciated in the rebuttal.
* Reviewer xYhQ asked about the insights behind using the Laplacian spectrum. In the rebuttal, the authors mentioned that it is a good characterization of graph structure and connectivity empirically. Reviewer 2dvj also asked if only using an aggregated scalar would be overly lossy. The authors' rebuttal is still only empirical. Neither reviewer showed clear satisfaction to the replies, and I also did not feel the replies could address my third bullet point.
* Reviewer xYhQ asked the situation of using a better, semi-empirical force field as the prior model. The authors showed the results and effectiveness in this situation in the rebuttal.
* Reviewer UBE3 asked about the unphysical discontinuity in the RR method. The authors' reply indicates that this discontinuity is present in popular GNNs and its artificial effect can be accepted. Nevertheless, this does not address my fourth bullet point.

After the rebuttal, Reviewer UBE3 increased his/her score considering the responsive more results and misunderstanding clarifications. Other reviewers remained hesitating around the borderline. Due to the remaining insufficiencies listed in the metareview, I tend to rate a reject.

---

### Decision · Program_Chairs · 2025-01-22

Reject